# Pre-trained Large Language Models Use Fourier Features to Compute Addition

**Tianyi Zhou   Deqing Fu   Vatsal Sharan   Robin Jia**
Department of Computer Science
University of Southern California
Los Angeles, CA 90089
{tzhou029,deqingfu,vsharan,robinjia}@usc.edu

## Abstract

Pre-trained large language models (LLMs) exhibit impressive mathematical reasoning capabilities, yet how they compute basic arithmetic, such as addition, remains unclear. This paper shows that pre-trained LLMs add numbers using Fourier features—dimensions in the hidden state that represent numbers via a set of features sparse in the frequency domain. Within the model, MLP and attention layers use Fourier features in complementary ways: MLP layers primarily approximate the magnitude of the answer using low-frequency features, while attention layers primarily perform modular addition (e.g., computing whether the answer is even or odd) using high-frequency features. Pre-training is crucial for this mechanism: models trained from scratch to add numbers only exploit low-frequency features, leading to lower accuracy. Introducing pre-trained token embeddings to a randomly initialized model rescues its performance. Overall, our analysis demonstrates that appropriate pre-trained representations (e.g., Fourier features) can unlock the ability of Transformers to learn precise mechanisms for algorithmic tasks.

## 1   Introduction

Mathematical problem solving has become a crucial task for evaluating the reasoning capabilities of large language models (LLMs) [20, 7, 23, 13]. While LLMs exhibit impressive mathematical abilities [34, 17, 1, 45, 40, 4, 11], it remains unclear how they perform even basic mathematical tasks. Do LLMs apply mathematical principles when solving math problems, or do they merely reproduce memorized patterns from the training data?

In this work, we unravel how pre-trained language models solve simple mathematical problems such as "Put together 15 and 93. Answer: ___". Prior work has studied how Transformers, the underlying architecture of LLMs, perform certain mathematical tasks. Most studies [5, 14, 42, 2, 12, 28, 18, 36] focus on Transformers with a limited number of layers or those trained from scratch; [19] analyzes how the pre-trained GPT-2-small performs the greater-than task. Our work focuses on a different task from prior interpretability work—integer addition—and shows that pre-trained LLMs learn distinct mechanisms from randomly initialized Transformers.

In §3, we show that pre-trained language models compute addition with Fourier features—dimensions in the hidden state that represent numbers via a set of features sparse in the frequency domain. First, we analyze the behavior of pre-trained LLMs on the addition task after fine-tuning, which leads to almost perfect accuracy on the task. Rather than merely memorizing answers from the training data, the models progressively compute the final answer layer by layer. Next, we analyze the contributions of individual model components using Logit Lens [3]. We observe that some components primarily *approximate* the answer—they promote all numbers close to the correct answer in magnitude—while other components primarily *classify* the answer modulo $m$ for various numbers $m$. Then,

38th Conference on Neural Information Processing Systems (NeurIPS 2024).

we use Fourier analysis to isolate features in the residual stream responsible for the low-frequency "approximation" and high-frequency "classification" subtasks. Identifying these features allows us to precisely ablate the ability of the model to perform either approximation or classification by applying a low-pass or high-pass filter, respectively, to the outputs of different model components. We find that MLP layers contribute primarily to approximation, whereas attention layers contribute primarily to classification.

In §4, we show that *pre-training* is crucial for learning this mechanism. The same network trained from scratch with random initialization not only shows no signs of Fourier features, but also has lower accuracy. We identify pre-trained token embeddings as a key source of inductive bias that help the pre-trained model learn a more precise mechanism for addition. Across the pre-trained token embeddings of many different pre-trained models, Fourier analysis uncovers large magnitudes of components with periods 2, 5, and 10. Introducing pre-trained token embeddings when training the model from scratch enables the model to achieve perfect test accuracy. Finally, we show that the same Fourier feature mechanism is present not only in models that were pre-trained and then fine-tuned, but also in frozen pre-trained LLMs when prompted with arithmetic problems.

Overall, our work provides a mechanistic perspective on how pre-trained LLMs compute addition through the lens of Fourier analysis. It not only broadens the scope from only investigating few-layer Transformers trained to fit a particular data distribution to understanding LLMs as a whole, but also hints at how pre-training can lead to more precise model capabilities.

## 2 Problem Setup

**Task and Dataset.** We constructed a synthetic addition dataset for fine-tuning and evaluation purposes. Each example involves adding two numbers $\leq 260$, chosen because the maximum number that can be represented by a single token in the GPT-2-XL tokenizer is 520. For each pair of numbers between 0 and 260, we randomly sample one of five natural language question templates and combine it with the two numbers. The dataset is shuffled and then split into training (80%), validation (10%), and test (10%) sets. More details are provided in Appendix F. In Appendix C.3, we show our that results generalize to a different dataset formatted with reverse Polish notation.

**Model.** Unless otherwise stated, all experiments focus on the pre-trained GPT-2-XL model that has been fine-tuned on our addition dataset. This model, which consists of 48 layers and approximately 1.5 billion parameters, learns the task almost perfectly, with an accuracy of 99.74% on the held-out test set. We examine other models in §4.2 and §4.3.

**Transformers.** We focus on decoder-only Transformer models [41], which process text sequentially, token by token, from left to right. Each layer $\ell$ in the Transformer has an attention module with output $\text{Attn}^{(\ell)}$ and an MLP module with output $\text{MLP}^{(\ell)}$. Their outputs are added together to create a continuous residual stream $h$ [9], meaning that the token representation accumulates all additive updates within the residual stream, with the representation $h^{(\ell)}$ in the $\ell$-th layer given by:

$$h^{(\ell)} = h^{(\ell-1)} + \text{Attn}^{(\ell)} + \text{MLP}^{(\ell)}. \tag{1}$$

The output embedding $W^U$ projects the residual stream to the space of the vocabulary; applying the softmax function then yields the model's prediction. We provide formal definitions in Appendix A.

## 3 Language Models Solve Addition with Fourier Features

In this section, we analyze the internal mechanisms of LLMs when solving addition tasks, employing a Fourier analysis framework. We first show that the model initially approximates the solution before iteratively converging to the correct answer (§3.1). We then show that the model refines its initial approximation by computing the exact answer modulo 2, 5, and 10, employing Fourier components of those same periods (§3.2). Finally, we demonstrate through targeted ablations that the identified Fourier components are causally important for the model's computational processes (§3.3). Specifically, we show that MLP layers primarily approximate the magnitude of the answer, using low-frequency features, while attention layers primarily perform modular addition using high-frequency components.

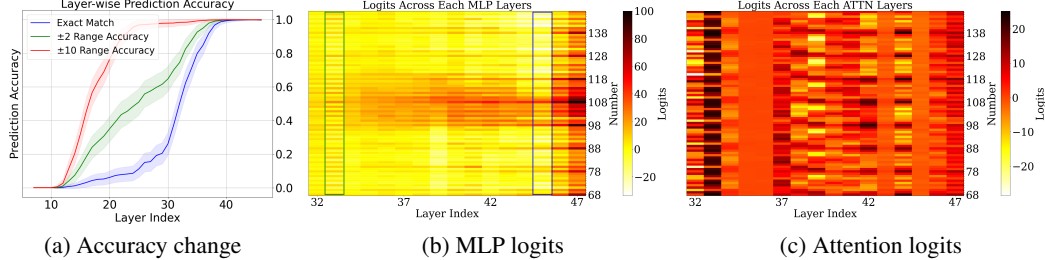

(a) Accuracy change          (b) MLP logits          (c) Attention logits

Figure 1: (a) Visualization of predictions extracted from fine-tuned GPT-2-XL at intermediate layers. Between layers 20 and 30, the model's accuracy is low, but its prediction is often within 10 of the correct answer: the model first approximates the answer, then refines it. (b) Heatmap of the logits from different MLP layers for the running example, "Put together 15 and 93. Answer: 108". The $y$-axis represents the subset of the number space around the correct prediction, while the $x$-axis represents the layer index. The 33-rd layer performs mod 2 operations (favoring even numbers), while other layers perform other modular addition operations, such as mod 10 (45-th layer). Additionally, most layers allocate more weight to numbers closer to the correct answer, 108. (c) Analogous plot for attention layers. Nearly all attention modules perform modular addition.

## 3.1 Behavioral Analysis

Our first goal is to understand whether the model merely memorizes and recombines pieces of information learned during training, or it performs calculations to add two numbers.

**Extracting intermediate predictions.** To elucidate how LLMs perform computations and progressively refine their outputs towards the correct answer, we extract model predictions at each layer from the residual stream. Let $L$ denote the number of layers. Using the Logit Lens method [3], instead of generating predictions by computing logits $W^U h^{(L)}$, predictions are derived through $W^U h^{(\ell)}$ where $\ell \in [L]$. We compute the accuracy of the prediction using each intermediate state $h^{(\ell)}$. If the models merely retrieve and recombine pieces of information learned during training, certain layers will directly map this information to predictions. For instance, [25] demonstrates that there is a specific MLP module directly maps a country to its capital.

**LLMs progressively compute the final answers.** Figure 1a instead shows that the model progressively approaches the correct answer, layer by layer. The model is capable of making predictions that fall within the range of $\pm 2$ and $\pm 10$ relative to the correct answer in the earlier layers, compared to the exact-match accuracy. This observation implies that the Transformer's layer-wise processing structure is beneficial for gradually refining predictions through a series of transformations and updates applied to the token representations.

## 3.2 Fourier Features in MLP & Attention Outputs

**Logits for MLP and attention have *periodic* structures.** We now analyze how each MLP and attention module contributes to the final prediction. We transform the output of the attention and MLP output at layer $\ell$ into the token space using $W^U \mathrm{Attn}^{(\ell)}$ and $W^U \mathrm{MLP}^{(\ell)}$ at each layer, thereby obtaining the logits $\mathcal{L}$ for each MLP and attention module. We use the running example "Put together 15 and 93. Answer: 108" to demonstrate how the fine-tuned GPT-2-XL performs the computation. As illustrated in Figure 1b and Figure 1c, both the MLP and attention modules exhibit a periodic pattern in their logits across the output number space, e.g., the MLP in layer 33, outlined in green, promotes all numbers that are congruent to 108 mod 2 (in Figure 20 in the appendix, we zoom into such layers to make this clearer). Overall, we observe two distinct types of computation within these components. Some components predominantly assign a high weight to numbers around the correct answer, which we term *approximation*. Meanwhile, other components predominantly assign a high weight to all numbers congruent to $a + b \bmod c$ for some constant $c$, which we term *classification*.

**Logits for MLP and attention are approximately sparse in the Fourier space.** It is natural to transform the logits into Fourier space to gain a better understanding of their properties such as the periodic pattern. We apply the discrete Fourier transform to represent the logits as the sum of

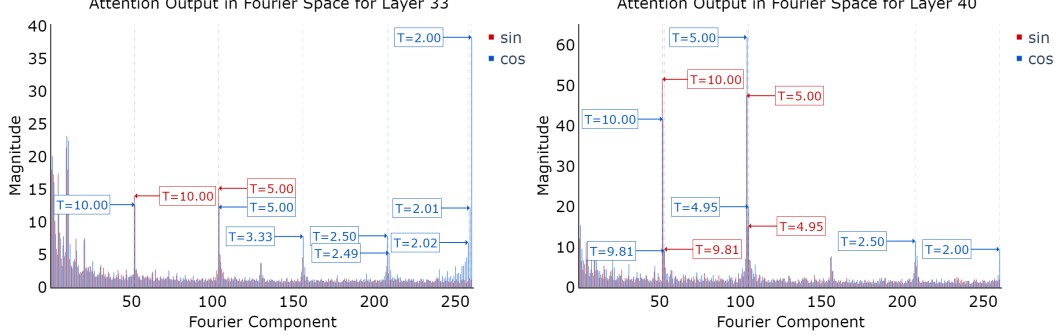

(a) Logits for the MLP output in Fourier space     (b) Logits for the attention output in Fourier space

Figure 2: The intermediate logits in Fourier space. We annotate the top-10 outlier high-frequency Fourier components based on their magnitudes. $T$ stands for the period of that Fourier component. (a) The logits in Fourier space for the MLP output of the 33-rd layer, i.e., $\widehat{\mathcal{L}}_{\mathrm{MLP}}^{(33)}$. The component with period 2 has the largest magnitude, aligning with the observations in Figures 1b and 20a. (b) The logits in Fourier space for the attention output of the 40-th layer, i.e., $\widehat{\mathcal{L}}_{\mathrm{Attn}}^{(40)}$. The components with periods 5 and 10 have the largest magnitude, aligning with the observations in Figures 1c and 20b.

sine and cosine waves of different periods: the $k$-th component in Fourier space has period $520/k$ and frequency $k/520$ (see Appendix A for more details). Let $\widehat{\mathcal{L}}$ denote the logits in Fourier space. Figure 2 shows the Fourier space logits for two layers from Figure 1b and Figure 1c that have a clear periodic pattern. We find that the high-frequency components in Fourier space, which we define as components with index greater or equal to 50, are approximately sparse as depicted in Figure 2. This observation aligns with [28], which found that a one-layer Transformer utilizes particular Fourier components within the Fourier space to solve the modular addition task.

In Figure 3, we show that similar sparsity patterns in Fourier space hold across the entire dataset. We compute the logits in Fourier space for the last 15 layers, i.e., $\widehat{\mathcal{L}}_{\mathrm{Attn}}^{(\ell)}$ and $\widehat{\mathcal{L}}_{\mathrm{MLP}}^{(\ell)}$ where $\ell \in [32, 47]$, for all test examples and average them. We annotate the top-10 outlier high-frequency Fourier components based on their magnitude. The MLPs also exhibit some strong low-frequency components; the attention modules do not exhibit strong low-frequency components, only high-frequency components.

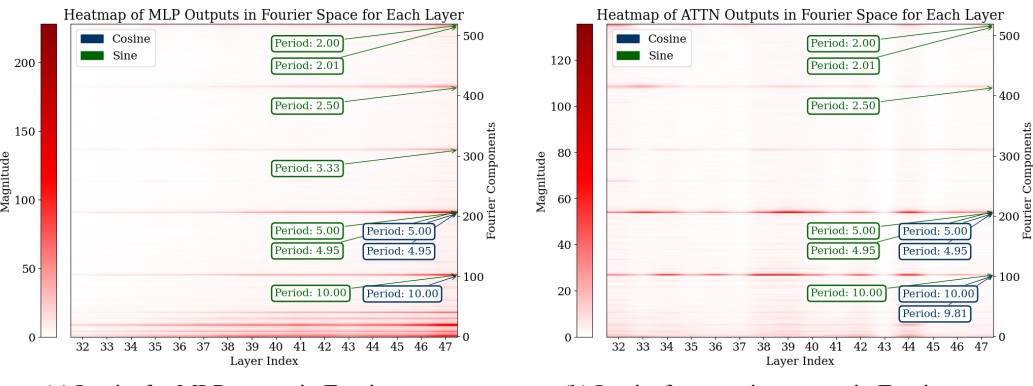

(a) Logits for MLP output in Fourier space     (b) Logits for attention output in Fourier space

Figure 3: Analysis of logits in Fourier space for all the test data across the last 15 layers. For both the MLP and attention modules, outlier Fourier components have periods around 2, 2.5, 5, and 10.

**Final logits are superpositions of these outlier Fourier components.** The final logits, $\mathcal{L}^{(L)}$, are the sum of all $\mathcal{L}_{\mathrm{MLP}}^{(l)}$ and $\mathcal{L}_{\mathrm{Attn}}^{(l)}$ across all layers $l \in [L]$. Figure 4 elucidates how these distinct Fourier components contribute to the final prediction, for the example "Put together 15 and 93. Answer: 108".

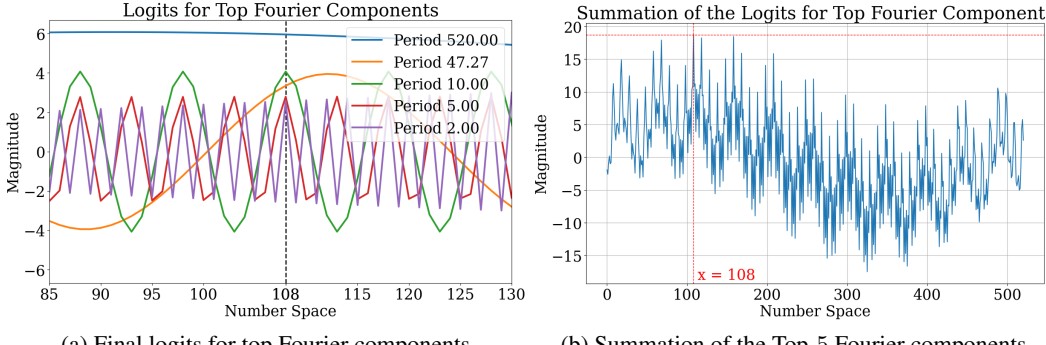

(a) Final logits for top Fourier components        (b) Summation of the Top-5 Fourier components

Figure 4: Visualization of how a sparse subset Fourier components can identify the correct answer. (a) Shows the top-5 Fourier components for the final logits. (b) Shows the sum of these top-5 Fourier components, highlighting how the cumulative effect identifies the correct answer, 108.

We select the top-5 Fourier components of $\widehat{\mathcal{L}}^{(L)}$ based on their magnitudes and transfer them back to logits in number space via the inverse discrete Fourier transform (Figure 4a). The large-period (low-frequency) components approximate the magnitude while the small-period (high-frequency) components are crucial for modular addition. Figure 4b shows that aggregating these 5 waves is sufficient to predict the correct answer.

**Why is high-frequency classification helpful?** The Fourier basis comprises both $\cos$ and $\sin$ waves (see Definition A.3). By adjusting the coefficients of $\cos$ and $\sin$, the trained model can manipulate the phase of the logits in Fourier space (number shift in number space), aligning the peak of the wave more closely with the correct answer. As shown in Figure 4a, consider a wave with a period of 2. Here, the peak occurs at every even number in the number space, corresponding to the mod 2 task. In contrast, for components with a large period such as 520, the model struggles to accurately position the peak at 108 (also see Figure 14 in the appendix for the plot of this component with period 520 in the full number space). This scenario can be interpreted as solving a "mod 520" task—a classification task among 520 classes—which is challenging for the model to learn accurately. Nevertheless, even though the component with a period of 520 does not solve the "mod 520" task precisely, it does succeed in assigning more weight to numbers near 108. The classification results from the high-frequency components can then provide finer-grained resolution to distinguish between all the numbers around 108 assigned a large weight by the lower frequencies. Due to this, the low-frequency components need not be perfectly aligned with the answer to make accurate predictions.

### 3.3 Fourier Features are Causally Important for Model Predictions

In the previous section, we demonstrated that there are outlier Fourier components in the logits generated by both the MLP and attention modules, as shown in Figure 3. We also illustrated that, in one example, the low-frequency components primarily approximate the magnitude, while the high-frequency components are crucial for modular addition tasks, as depicted in Figure 4. In this section, through an ablation study conducted across the entire test dataset, we show that both types of components are essential for correctly computing sums. Moreover, we reveal that the MLP layers primarily approximate the magnitude of the answer using low-frequency features, whereas the attention layers are responsible for modular addition using high-frequency features.

**Filtering out Fourier components.** To understand the role various frequency components play for the addition task, we introduce low-pass and high-pass filters $\mathcal{F}$. For an intermediate state $h$, and a set of frequencies $\Gamma = \{\gamma_1, \ldots, \gamma_k\}$, the filter $\mathcal{F}(h; \Gamma)$ returns the vector $\widetilde{h}$ that is closest in $L_2$ distance to $h$ subject to the constraint that the Fourier decomposition of $W^U \widetilde{h}$ at every frequency $\gamma_i$ is 0. We show in Appendix A that this has a simple closed-form solution involving a linear projection. We then apply either a low-pass filter by taking $\Gamma$ to be all the components whose frequencies are greater than the frequency of the $\tau$-th component for some threshold $\tau$ (i.e., removing high-frequency components), and a high-pass filter by taking $\Gamma$ to be all the components whose frequencies are less

Table 1: Impact of Filtering out Fourier Components on Model Performance. Removing low-frequency components from attention modules (blue) or high-frequency components from MLP modules (red) does not impact performance

| Module | Fourier Component Removed | Validation Loss | Accuracy |
|---|---|---|---|
| None | Without Filtering | 0.0073 | 0.9974 |
| ATTN & MLP | Low-Frequency | 4.0842 | 0.0594 |
| ATTN | Low-Frequency | 0.0352 | 0.9912 |
| MLP | Low-Frequency | 2.1399 | 0.3589 |
| ATTN & MLP | High-Frequency | 1.8598 | 0.2708 |
| ATTN | High-Frequency | 0.5943 | 0.7836 |
| MLP | High-Frequency | 0.1213 | 0.9810 |

than the frequency of the $\tau$-th component (i.e., removing low-frequency components). As in the previous subsection, we take the high-frequency threshold $\tau = 50$ for the following experiments (see Appendix B for more details).

**Different roles of frequency components in approximation and classification tasks.** We evaluated the fine-tuned GPT-2-XL model on the test dataset with different frequency filters applied to all of the output of MLP and attention modules. The results, presented in Table 1, indicate that removing low-frequency components from attention modules or high-frequency components from MLP modules does not impact performance. This observation suggests that attention modules are not crucial for approximation tasks, and MLP modules are less significant for classification tasks.

Eliminating high-frequency components from attention results in a noticeable decrease in accuracy. Furthermore, removing high-frequency components from both the attention and MLP modules simultaneously leads to an even greater reduction in accuracy. This finding corresponds with observations from Figure 1b,c and Figure 3, which indicate that both MLP and attention modules are involved in classification tasks due to the presence of high-frequency components in the logits. As shown in Table 1, the approximation tasks are primarily performed by the MLP modules, with contributions from the attention modules as well.

The errors induced by these ablations align with our mechanistic understanding. Ablating low-frequency parts of MLPs leads to off-by 10, 50, and 100 errors: the model fails to perform the approximation subtask, though it still accurately predicts the unit digit. Conversely, ablating high-frequency parts of attention leads to small errors less than 6 in magnitude: the model struggles to accurately predict the units digit, but it can still estimate the overall magnitude of the answer. See Figure 21 in the Appendix for more details. These observations validate our hypothesis that low-frequency components are crucial for approximation, while high-frequency components are vital for classification. The primary function of MLP modules is to approximate the magnitude of outcomes using low-frequency components, while the primary role of attention modules is to ensure accurate classification by determining the correct unit digit.

## 4 Effects of Pre-training

The previous section shows that pre-trained LLMs leverage Fourier features to solve the addition problem. Now, we study where the models' reliance on Fourier features comes from. In this section, we demonstrate that LLMs learn Fourier features in the token embeddings for numbers during pre-training. These token embeddings are important for achieving high accuracy on the addition task: models trained from scratch achieve lower accuracy, but adding just the pre-trained token embeddings fixes this problem. We also show that pre-trained models leverage Fourier features not only when fine-tuned, but also when prompted.

### 4.1 Fourier features in Token Embedding

**Number embedding exhibits approximate sparsity in the Fourier space.** Let $W^E \in \mathbb{R}^{p \times D}$, where $p = 521$ and $D$ is the size of the token embeddings, denote the token embedding for numbers. We apply the discrete Fourier transform to each column of $W^E$ to obtain a matrix $V \in \mathbb{R}^{p \times D}$, where

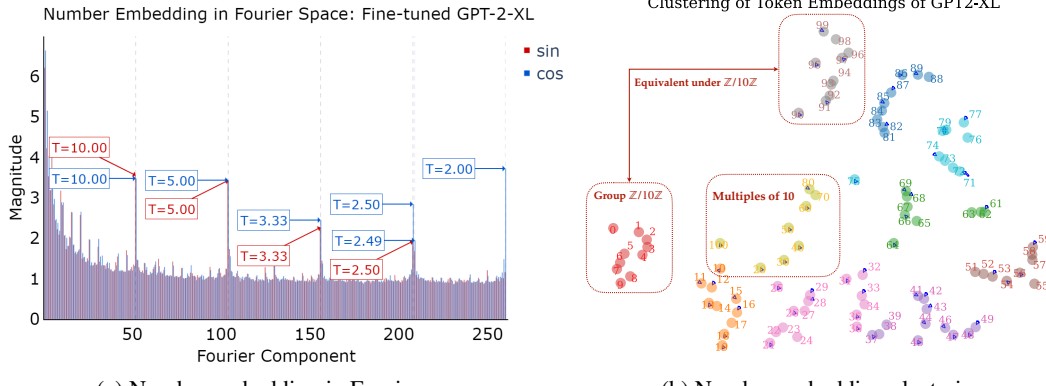

(a) Number embedding in Fourier space

(b) Number embedding clustering

Figure 5: (a) Number embedding in Fourier space for fine-tuned GPT-2-XL. $T$ stands for the period of that Fourier component.(b) Visualization of token embedding clustering of GPT-2 using T-SNE and $k$-means with 10 clusters. The numbers are clustered based on their magnitude and whether they are multiples of 10.

each row represents a different Fourier component. Then we take the $L_2$ norm of each row to yield a $p$-dimensional vector. Each component $j$ in this vector measures the overall magnitude of the $j$-th Fourier component across all the token embedding dimensions. Figure 5a shows the magnitude of different Fourier components in the token embedding of GPT-2-XL. We see that the token embedding has outlier components whose periods are $2, 2.5, 5$, and $10$. Therefore, similar to how the model uses different Fourier components to represent its prediction (as shown in Section 3.2), the token embeddings represent numbers with different Fourier components. Figure 15 in the Appendix shows that the token embeddings of other pre-trained models have similar patterns the Fourier space. This suggests that Fourier features are a common attribute in the token embedding of pre-trained LLMs. In Figure 5b, we use t-SNE and $k$-means to visualize the token embedding clustering. We can see that numbers cluster not only by magnitude but also by their multiples of 10.

## 4.2 Contrasting Pre-trained Models with Models Trained from Scratch

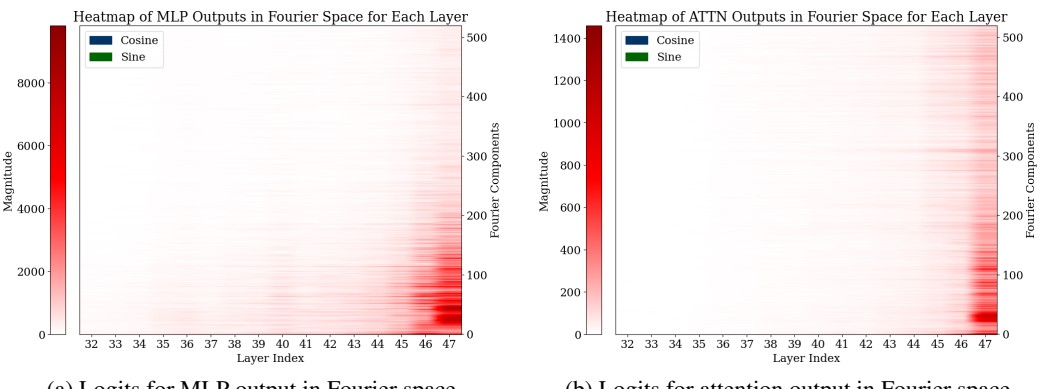

(a) Logits for MLP output in Fourier space

(b) Logits for attention output in Fourier space

Figure 6: Visualization of the logits in Fourier space on the test dataset from the last 15 layers for the GPT-2-XL model *trained from scratch*. For both the MLP and attention modules, there are no outlier Fourier components, in contrast with the clear outlier components in the fine-tuned model (Figure 3).

To understand the necessity of Fourier features for the addition problem, we trained the GPT-2-XL model from scratch on the addition task with random initialization. After convergence, it achieved only $94.44\%$ test accuracy (recall that the fine-tuned GPT-2-XL model achieved $99.74\%$ accuracy).

**Fourier features are learned during pre-training.** Figure 6 shows that there are no Fourier features in the intermediate logits of the GPT-2-XL model trained from scratch on the addition task. Furthermore, Figure 7a shows that the token embeddings also have no Fourier features. Without leveraging Fourier features, the model merely approximates the correct answer without performing modular addition, resulting in frequent off-by-one errors between the prediction and the correct answer (see details in Figure 23).

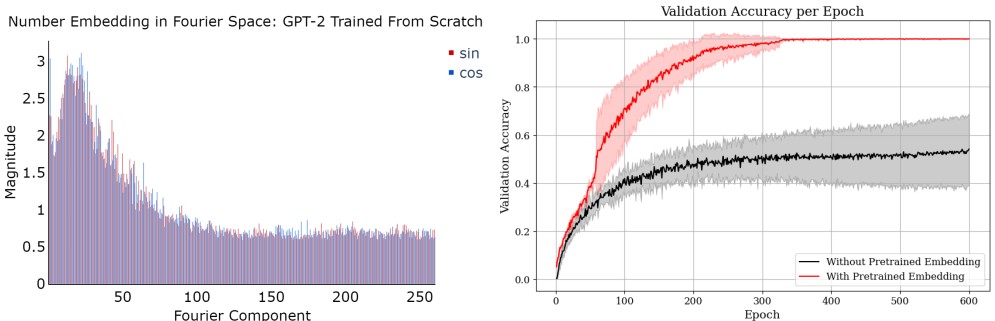

(a) Embedding: GPT-2-XL Trained from Scratch      (b) Validation Accuracy Comparison for GPT-2

Figure 7: (a) The number embedding in Fourier space for GPT-2-XL trained from scratch. There are no high-frequency outlier components, in contrast with the pre-trained embeddings (Figure 5a). (b) Validation accuracy of GPT-2-small trained from scratch either with or without pre-trained token embeddings. We show the mean and the standard deviation of the validation accuracy across 5 random seeds. GPT-2-small with pre-trained token embedding consistently achieves 100% accuracy, while GPT-2-small without pre-trained token embedding only achieves less than 60% accuracy.

**Pre-trained token embeddings improve model training.** We also trained GPT-2-small, with 124 million parameters and 12 layers, from scratch on the addition task. GPT-2-small often struggles with mathematical tasks [26]. This model achieved a test accuracy of only 53.95% after convergence. However, when we freeze the token embedding layer and randomly initialize the weights for all other layers before training on the addition task, the test accuracy increases to 100%, with a significantly faster convergence rate. This outcome was consistently observed across five different random seeds, as illustrated in Figure 7b. Following Section 3.2, to validate that the model learn to leverage the Fourier feature to solve addition, we analyze the logits in Fourier space for all the test data across all 12 layers In Figure 8, we can clearly observe that, with solely the pre-trained number embedding, the Fourier features appear in the MLP and attention modules' output for most layers. This demonstrates that given the number embeddings with Fourier features, the model can effectively learn to leverage these features to solve the addition task.

### 4.3 Fourier Features in Prompted Pre-Trained Models

Finally, we ask whether larger language models use similar Fourier features during prompting.

**Pre-trained LLMs use Fourier features to compute addition during in-context learning.** We first test on the open-source models GPT-J [43] with 6B parameters, and Phi-2 [21] with 2.7B parameters on the test dataset. Without in-context learning, the model cannot perform addition tasks. Therefore, we use 4-shot in-context learning to test its performance. Their absolute errors are predominantly multiples of 10: 93% of the time for GPT-J, and 73% for Phi-2 . Using the Fourier analysis framework proposed in Section 3.2, we demonstrate that for Phi-2 and GPT-J, the outputs of MLP and attention modules exhibit approximate sparsity in Fourier space across the last 15 layers (Figure 9 and Figure 19). This evidence strongly suggests that these models leverage Fourier features to compute additions.

**Closed-source models exhibit similar behavior.** We study the closed-source models GPT-3.5 [33], GPT-4 [34], and PaLM-2 [16]. While we cannot analyze their internal representations, we can study whether their behavior on addition problems is consistent with reliance on Fourier features. Since closed-source LLMs are instruction tuned and perform well without in-context learning, we conduct error analysis with 0-shot. Most absolute errors by these models are also multiples of 10: 100% of the time for GPT-3.5 and GPT-4, and 87% for PaLM-2. The similarity in error distribution to

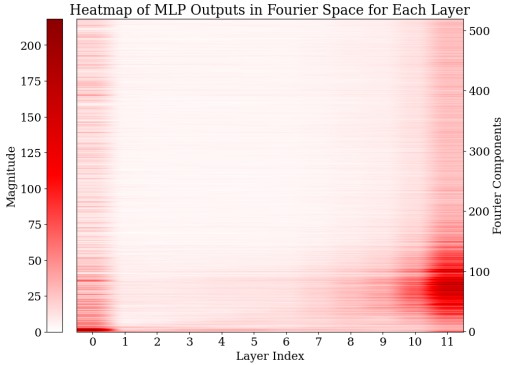

(a) Logits for MLP output in Fourier space **without** pre-trained number embeddings

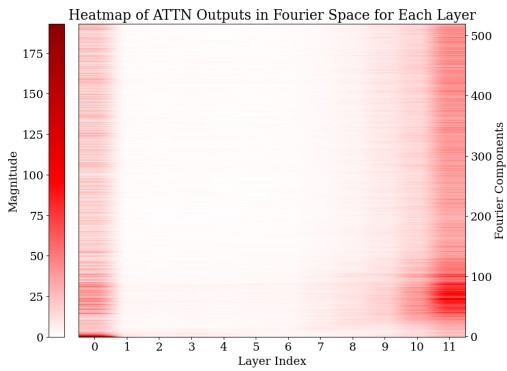

(b) Logits for attention output in Fourier space **without** pre-trained number embeddings

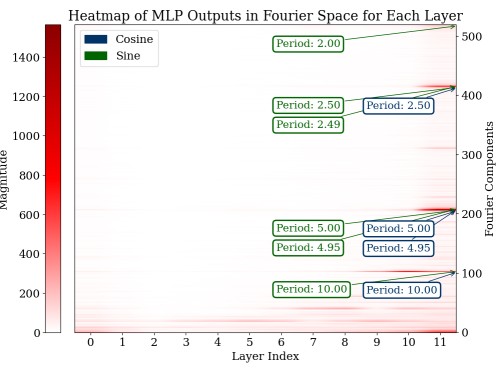

(c) Logits for MLP output in Fourier space **with** pre-trained number embeddings

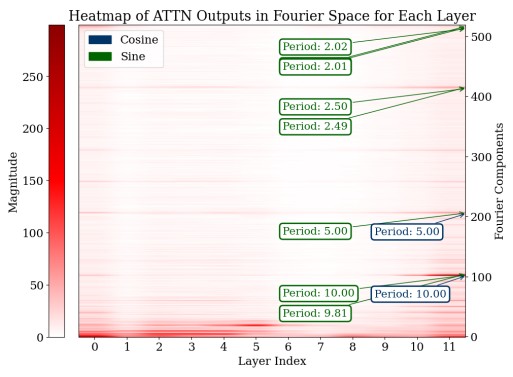

(d) Logits for attention output in Fourier space **with** pre-trained number embeddings

Figure 8: Analysis of logits in Fourier space for all the test data across the 12 layers. **(a,b)** GPT-2-small trained from scratch fail to learn to leverage the Fourier feature to solve addition. **(c,d)** However, with solely the pre-trained number embeddings, GPT-2-small is able to learn to leverage the Fourier features to solve the addition as the fine-tuned models.

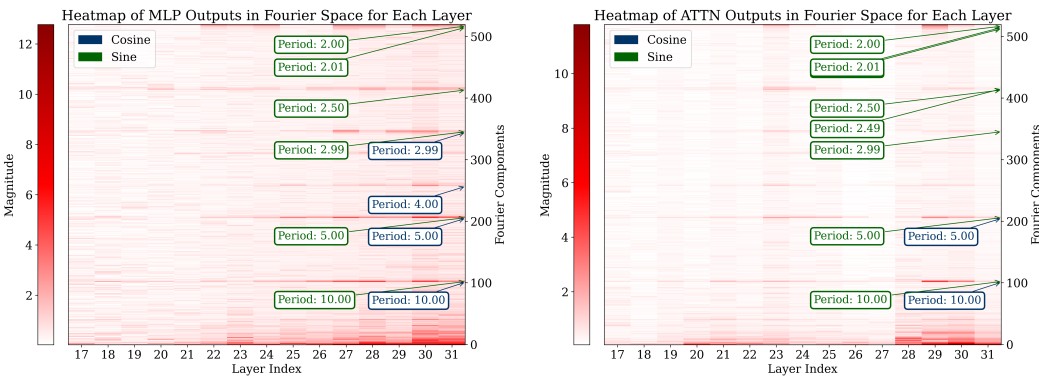

(a) Logits for MLP output in Fourier space          (b) Logits for attention output in Fourier space

Figure 9: For Phi-2 (4-shot), we analyzed the logits in Fourier space for all the test data across the last 15 layers. For both the MLP and attention modules, the outlier Fourier components have periods around 2, 2.5, 5, and 10, similar to the fine-tuned GPT-2-XL logits (Figure 3).

that of open-source models leads us to hypothesize that Fourier features play a critical role in their computational mechanism.

# 5 Related Work

**Learning mathematical tasks.** Previous studies primarily explore what pre-trained LMs can achieve on arithmetic tasks, with less emphasis on the underlying mechanisms [29, 37]. For instance, [22] demonstrates that small Transformer models can effectively learn arithmetic by altering the question format and utilizing a scratchpad method [30]. [19] identifies activation patterns for the "greater-than" operation in GPT-2, and [5] focuses on the enumeration and selection processes in GCD computation. In this paper, we dive into the specific roles of MLP and attention layers in solving mathematical tasks. Our research analyzes these components' distinct contributions to integer addition tasks.

**Mechanisms of pre-trained LMs.** Recent studies have significantly advanced our understanding of the underlying mechanisms of pre-trained Transformer models. For instance, research on "skill neurons" by [44] and "knowledge neurons" by [8] underscores the development of specialized neural components that encode task-specific capabilities or hold explicit factual information in the pre-trained LMs, enhancing model performance on related tasks. [25] and [15] discuss how MLPs and FFNs transform and update token representations for general language tasks. In contrast, we show that the pre-trained LMs use multiple layers to compute addition by combining the results of approximation and classification. Additionally, [46] demonstrated the capacity of GPT-2 to consolidate similar information through pre-training in the model weights, which aligns with our observations on the importance of pre-training in developing effective number embedding and arithmetic computation strategies in LMs.

**Fourier features in Neural Networks.** Fourier features are commonly observed in image models, particularly in the early layers of vision models [32, 31, 10]. These features enable the model to detect edges, textures, and other spatial patterns effectively. Recently, Fourier features have been noted in networks trained for tasks that allow cyclic wraparound, such as modular addition [28, 27], general group compositions [6], or invariance to cyclic translations [38]. [28] demonstrates that learning Fourier features can induce 'grokking' [35]. Furthermore, [24] provides a mathematical framework explaining the emergence of Fourier features when the network exhibits invariance to a finite group. We extend these insights by observing Fourier features in tasks that do not involve cyclic wraparound. [39] found that by selecting problem-specific Fourier features, the performance of MLPs can be improved on a computer vision-related task.

# 6 Conclusion

In this paper, we provide a comprehensive analysis of how pre-trained LLMs compute numerical sums, revealing a nuanced interplay of Fourier features within their architecture. Our findings demonstrate that LLMs do not simply memorize answers from training data but actively compute solutions through a combination of approximation and classification processes encoded in the frequency domain of their hidden states. Specifically, MLP layers contribute to approximating the magnitude of sums, while attention layers contribute to modular operations.

Our work also shows that pre-training plays a critical role in equipping LLMs with the Fourier features necessary for executing arithmetic operations. Models trained from scratch lack these crucial features and achieve lower accuracy; introducing pre-trained token embeddings greatly improves their convergence rate and accuracy. This insight into the arithmetic problem-solving capabilities of LLMs through Fourier features sets the stage for potential modifications to training approaches. By imposing specific constraints on model training, we could further enhance the ability of LLMs to learn and leverage these Fourier features, thereby improving their performance in mathematical tasks.

# 7 Limitations

We note that our contributions are limited by the size of the dataset. As the maximum number that can be represented by one token for GPT-2-XL is 520, we analyze on the dataset whose operands are less than 260. However, as the Fourier features commonly exist in many different pre-trained models as shown in Section 4, we believe different models still use Fourier features, possibly with a more complicated strategy.

## Acknowledgments

DF and RJ were supported by a Google Research Scholar Award. RJ was also supported by an Open Philanthropy research grant. VS was supported by NSF CAREER Award CCF-2239265 and an Amazon Research Award. Any opinions, findings, and conclusions or recommendations expressed in this material are those of the author(s) and do not reflect the views of the funding agencies.

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

# Appendix

**Roadmap.** In Appendix A, we introduce some formal definitions that used in our main content. In Appendix B, we show why we separate the Fourier components into the high-frequency part and the low-frequency part and why we choose $\tau$ to be 50. In Appendix C, we show our observation generalizes to another format of dataset, another arithmetic task and other models. In Appendix D, we provide more evidence that shows the Fourier features in the model when computing addition. In Appendix E, we provide more evidence that shows the GPT-2-XL trained from scratch does not use Fourier feature to solve the addition task. In Appendix F, we give the details of our experimental settings.

## A    Formal Definition of Transformer and Logits in Fourier Space

We first introduce the formal definition of the Transformer structure that we used in this paper.

**Definition A.1** (Transformer). *An autoregressive Transformer language model $G : \mathcal{X} \to \mathcal{Y}$ over vocabulary* Vocab *maps a token sequence $x = [x_1, \ldots, x_N] \in \mathcal{X}, x_t \in$ Vocab *to a probability distribution $y \in \mathcal{Y} \subset \mathbb{R}^{|\text{Vocab}|}$ that predicts next-token continuations of $x$. Within the Transformer, the $i$-th token is embedded as a series of hidden state vectors $h_t^{(\ell)}$, beginning with $h_t^{(0)} = \text{emb}(x_t) + \text{pos}(i) \in \mathbb{R}^D$. Let $W^U \in \mathbb{R}^{|\text{Vocab}| \times D}$ denote the output embedding. The final output $y = \text{softmax}(W^U\left(h_N^{(L)}\right))$ is read from the last hidden state. In the autoregressive case, tokens only draw information from past tokens:*

$$h_t^{(\ell)} = h_t^{(\ell-1)} + \text{Attn}_t^{(\ell)} + \text{MLP}_t^{(\ell)}$$

*where*

$$\text{Attn}_t^{(\ell)} := \text{Attn}^{(\ell)}\left(h_1^{(\ell-1)}, h_2^{(\ell-1)}, \ldots, h_t^{(\ell-1)}\right) \quad \text{and} \quad \text{MLP}_t^{(\ell)} := \text{MLP}_t^{(\ell)}(\text{Attn}_t^{(\ell)}, h_t^{(\ell-1)}).$$

In this paper, we only consider the output tokens to be numbers. Hence, we have the unembedding matrix $W^U \in \mathbb{R}^{p \times D}$, where $p$ is the size of the number space. As we are given the length-$N$ input sequences and predict the $(N+1)$-th, we only consider $h_N^{(\ell)} = h_N^{(\ell-1)} + \text{Attn}_N^{(\ell)} + \text{MLP}_N^{(\ell)}$. For simplicity, we ignore the subscript $N$ in the following paper, so we get Eq. (1).

**Definition A.2** (Intermediate Logits). *Let $\mathcal{L}_{\text{Attn}}^{(\ell)} := W^U \text{Attn}^{(\ell)}$ denote the intermediate logits of the attention module at the $\ell$-th layer. Let $\mathcal{L}_{\text{MLP}}^{(\ell)} := W^U \text{MLP}^{(\ell)}$ denote the intermediate logits of the MLP module at the $\ell$-th layer. Let $\mathcal{L}^{(\ell)} := W^U h^{(\ell)}$ denote the logits on intermediate state $h^{(\ell)}$.*

Throughout the model, $h$ undergoes only additive updates (Eq. (1)), creating a continuous residual stream [9], meaning that the token representation $h$ accumulates all additive updates within the residual stream up to layer $t$.

To analyze the logits in Fourier space, we give the formal definition of the Fourier basis as follows:

**Definition A.3** (Fourier Basis). *Let $p$ denote the size of the number space. Let $\overrightarrow{\mathbf{x}} := (0, 1, \ldots, (p-1))$. Let $\omega_k := \frac{2\pi k}{p-1}$. We denote the normalized Fourier basis $F$ as the $p \times p$ matrix:*

$$F := \begin{bmatrix} \sqrt{\frac{1}{p-1}} \cdot \overrightarrow{\mathbf{1}} \\ \sqrt{\frac{2}{p-1}} \cdot \sin\left(\omega_1 \overrightarrow{\mathbf{x}}\right) \\ \sqrt{\frac{2}{p-1}} \cdot \cos\left(\omega_1 \overrightarrow{\mathbf{x}}\right) \\ \sqrt{\frac{2}{p-1}} \cdot \sin\left(\omega_2 \overrightarrow{\mathbf{x}}\right) \\ \vdots \\ \sqrt{\frac{2}{p-1}} \cdot \cos\left(\omega_{(p-1)/2} \overrightarrow{\mathbf{x}}\right) \end{bmatrix} \in \mathbb{R}^{p \times p}$$

*The first component $F[0]$ is defined as a constant component. For $i \in [0, p-1]$, $F[i]$ is defined as the $k$-th component in Fourier space, where $k = \lfloor \frac{i+1}{2} \rfloor$. The frequency of the $k$-th component is $f_k := \frac{k}{p-1}$. The period of the $k$-th component is $T_k := \frac{p-1}{k}$*

We can compute the discrete Fourier transform under that Fourier basis as follows:

**Remark A.4** (Discrete Fourier transformer (DFT) and inverse DFT). *We can transform any logits $u \in \mathbb{R}^p$ to Fourier space by computing $\widehat{u} = F \cdot u$. We can transform $\widehat{u}$ back to $u$ by $u = F^\top \cdot \widehat{u}$*

Next, we define the logits in Fourier space.

**Definition A.5** (Logits in Fourier Space). *Let $\mathcal{L}^{(L)}$, $\mathcal{L}^{(\ell)}_{\text{Attn}}$ and $\mathcal{L}^{(\ell)}_{\text{MLP}}$ denote the logits (Definition A.2). The output logits before softmax in Fourier space is defined as: $\widehat{\mathcal{L}}^{(L)} = F \cdot \mathcal{L}^{(L)}$. The logits of the MLP and attention modules in Fourier space are defined as:*

$$\widehat{\mathcal{L}}^{(\ell)}_{\text{Attn}} = F \cdot \mathcal{L}^{(\ell)}_{\text{Attn}} \quad \text{and} \quad \widehat{\mathcal{L}}^{(\ell)}_{\text{MLP}} = F \cdot \mathcal{L}^{(\ell)}_{\text{MLP}}.$$

We ignore the first elements in $\widehat{\mathcal{L}}^{(L)}, \widehat{\mathcal{L}}^{(\ell)}_{\text{Attn}}$ and $\widehat{\mathcal{L}}^{(\ell)}_{\text{MLP}}$ for the Fourier analysis in this paper as they are the constant terms. Adding a constant to the logits will not change the prediction.

Let $\tau \in \mathbb{R}$ denote a constant threshold. The low-frequency components for the logits in Fourier space are defined as $\widehat{\mathcal{L}}^{(\ell)}[1 : 2\tau]$. The high-frequency components for the logits in Fourier space are defined as $\widehat{\mathcal{L}}^{(\ell)}[2\tau :]$. For the following analysis, we choose $\tau = 50$ (the specific choice of $\tau = 50$ is explained in Appendix B).

Next, we propose the formal definition of low-pass/high-pass filter that is used in the following ablation study.

**Definition A.6** (Loss-pass / High-pass Filter). *Let $x \in \mathbb{R}^D$ denote the output of MLP or attention modules. Let $F$ denote the Fourier Basis (Definition A.3). Let $\tau \in R$ denote the frequency threshold. Let $W^U \in R^{p \times D}$ denote the output embedding. For low-pass filter, we define a diagonal binary matrix $B \in \{0, 1\}^{p \times p}$ as $b_{ii} = \begin{cases} 1 & \text{if } i \geq \tau \\ 0 & \text{otherwise} \end{cases}$. For high-pass filter, we define a diagonal binary matrix $B \in \{0, 1\}^{p \times p}$ as $b_{ii} = \begin{cases} 1 & \text{if } 1 \leq i < \tau \\ 0 & \text{otherwise} \end{cases}$. Note that we retain the constant component, so $b_{i,i} = 0$. The output of the filter $\mathcal{F}(x) : \mathbb{R}^D \to \mathbb{R}^D$ is defined by the following objective function:*

$$\min_y \quad \|x - y\|_2^2$$
$$\text{subject to} \quad BFW^U y = 0$$

The solution to the above optimization problem is given by a linear projection.

**Remark A.7.** *The result of the optimization problem defined in Definition A.6 is the projection of $x$ to the null space of $BFW^U$. Let $\mathcal{N}(BFW^U)$ denote the null space of $BFW^U$. We have*

$$\mathcal{F}(x) = \mathcal{N}(BFW^U) \cdot \mathcal{N}(BFW^U)^\top \cdot x^\top$$

# B  Fourier Components Separation and Selection of $\tau$

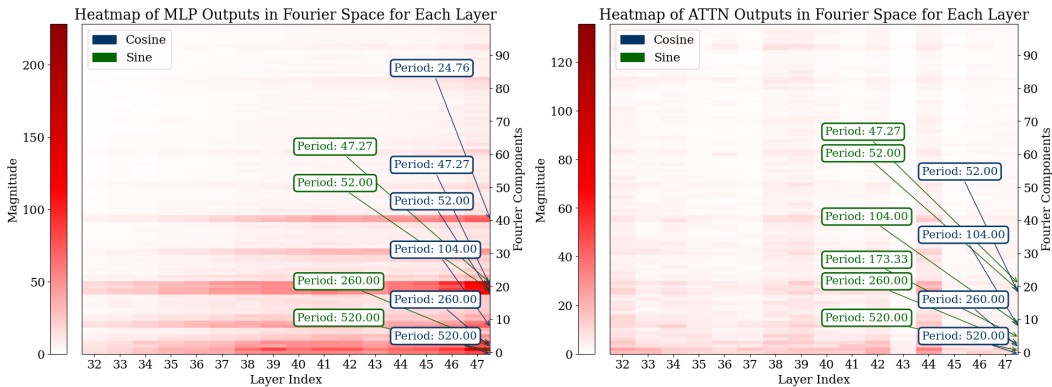

(a) Logits for MLP output in Fourier space    (b) Logits for attention output in Fourier space

Figure 10: We analyzed the logits in Fourier space for all the test data across the last 15 layers. For both the MLP and attention modules. We only plot the first 50 Fourier components (a) The MLP exhibits some outlier low-frequency Fourier components. (b) The attention module's low-frequency Fourier components are not as obvious as the ones in MLP.

Following Definition A.6, we define single-pass filter as follows:

**Definition B.1** (Single-Pass Filter). *Let $x \in \mathbb{R}^D$ denote the output of MLP or attention modules. Let $F$ denote the Fourier Basis (Definition A.3). Let $\gamma \in R$ denote the $\gamma$-th Fourier component (Definition A.3) that we want to retain. Let $W^U \in R^{V \times D}$ denote the output embedding. We define a diagonal binary matrix $B \in \{0,1\}^{V \times V}$ as $b_{ii} = \begin{cases} 0 & \text{if } \lfloor \frac{i+1}{2} \rfloor = \gamma \text{ or } i = 0, \\ 1 & \text{otherwise}. \end{cases}$*

*The output of the filter $\mathcal{F}_\gamma(x) : \mathbb{R}^D \to \mathbb{R}^D$ is defined as the following objective function:*

$$\min_y \quad \|x - y\|_2^2$$

$$\text{subject to} \quad BFW^U y = 0$$

**Remark B.2.** *The result of the optimization problem defined in Definition B.1 is the projection of $x$ to the null space of $BFW^U$. Let $\mathcal{N}(BFW^U)$ denote the null space of $BFW^U$. We have*

$$\mathcal{F}_\gamma(x) = \mathcal{N}(BFW^U) \cdot \mathcal{N}(BFW^U)^\top \cdot x^\top$$

For the single-pass filter, we only retrain one Fourier component and analyze how this component affects the model's prediction. The residual stream is then updated as follows:

$$h^{(\ell)} = h^{(\ell-1)} + \mathcal{F}_\gamma(\text{Attn}^{(\ell-1)}) + \mathcal{F}_\gamma(\text{MLP}^{(\ell-1)})$$

We evaluated the fine-tuned GPT-2-XL model on the addition dataset with the Fourier components period 520 and 2. Given that $T_k := \frac{V-1}{k}$ (Definition A.3), we retained only the Fourier components with $\gamma = 1$ and 260, respectively.

As shown in Figure 11a, with only one frequency component, whose period is 2, the model accurately predicts the parity with 99.59% accuracy. As depicted in Figure 11b, with a single frequency component of period 520, the model fails to accurately predict with 96.51% accuracy. We consider the frequency component with a period of 2 as the model's prediction for the `mod` 2 task, and the frequency component with a period of 520 as its prediction for the `mod` 520 task. Figures 11 and 12 suggest that the model effectively learns the `mod` 2 task, as it involves a two-class classification, but struggles with the `mod` 520 task, which requires classifying among 520 classes. As the model does not need to be trained to converge to the optimal for these low-frequency components as explained at the end of Section 3.2, predicting with the period-520 component leads to predictions that normally distributed around the correct answers.

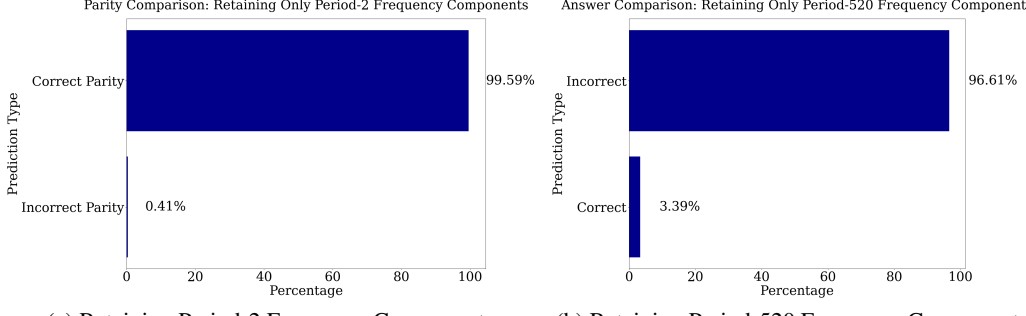

(a) Retaining Period-2 Frequency Components      (b) Retaining Period-520 Frequency Components

Figure 11: The prediction analysis when predicting with only one Fourier component. (a) Retaining only the Period-2 Fourier component makes the prediction $99.59\%$ accurate for `mod 2` task (b) Retaining only the Period-520 Fourier component makes the prediction $96.61\%$ inaccurate for the `mod 520` task.

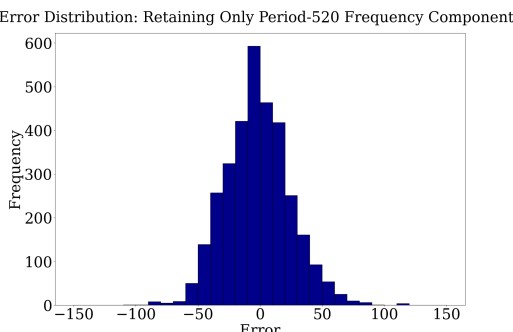

Figure 12: Retaining only the Period-520 Fourier component makes the model's predictions normally distributed around the correct answers.

The Fourier components with larger periods present greater difficulty in solving the corresponding modular addition task compared to those with smaller periods. As demonstrated in Figure 12, components with large periods serve primarily as approximations of the correct answer. Consequently, we categorize the Fourier components into low-frequency and high-frequency groups. The low-frequency components approximate the magnitude of the answer, whereas the high-frequency components are employed to enhance the precision of the predictions.

In reference to Figure 4, to elucidate the contribution of these distinct Fourier components to our final prediction and the rationale behind their separation, consider the example: "Put together 15 and 93. Answer: 108". We selected the top-10 Fourier components of $\widehat{\mathcal{L}}^{(L)}$ based on their magnitudes and converted them back to logits in the numerical space by multiplying with $F^\top$. We plotted the components with components index less than 50 in Figure 13a and those with components index greater than 50 in Figure 13b. Leveraging the constructive and destructive inference for different waves, the components with low periods assign more weight to the correct answer, 108, and less weight to numbers close to 108. These high-frequency (low-period) components ensure the prediction's accuracy at the unit place. For the low-frequency (large-period) components, the model fails to precisely learn the magnitude of the factor between the $\cos$ and $\sin$ components, which results in failing to peak at the correct answer. Thus, the low-frequency (large-period) components are used to approximate the magnitude of the addition results.

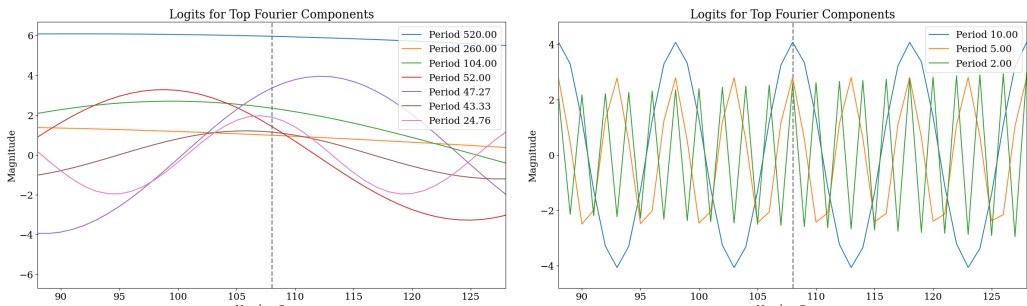

(a) Final Logits for Components whose index Less than 50

(b) Final Logits for Components whose index Greater than 50

Figure 13: Visualization of tthe individual final logits for the top-10 Fourier components with example "Put together 15 and 93. Answer: 108' (a) The components index less than 50. (b) The components index greater than 50.

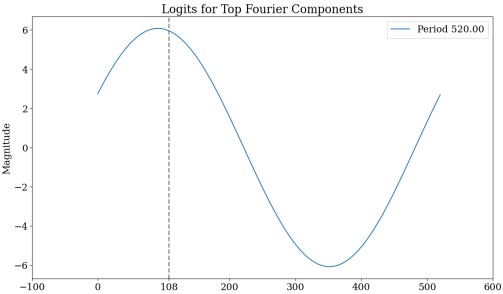

Figure 14: Visualization of the Fourier component whose period is 520 analysis for the final logits.

# C  Does Fourier Features Generalize?

## C.1  Token Embedding for Other LMs

We first show that other pre-trained LMs also have Fourier features in their token embedding for the numbers $[0, 520]$.

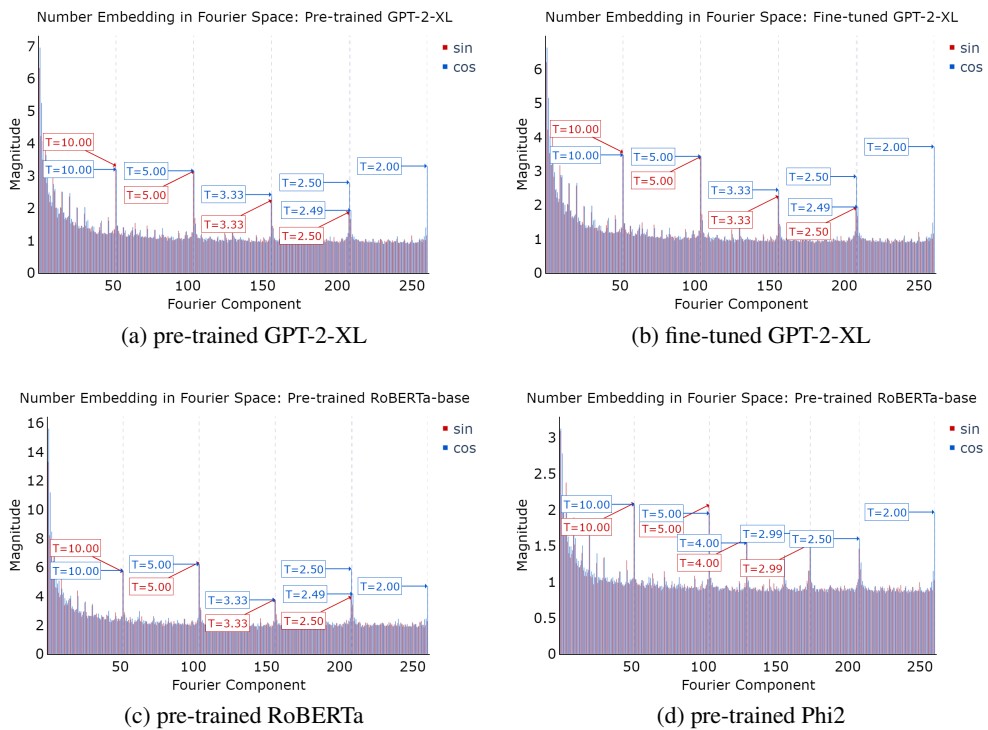

(a) pre-trained GPT-2-XL

(b) fine-tuned GPT-2-XL

(c) pre-trained RoBERTa

(d) pre-trained Phi2

Figure 15: Number embedding in Fourier space for different pre-trained models.

## C.2 Multiplication Task

A key question is whether pre-trained models utilize Fourier Features solely for solving addition tasks or if they generalize to other arithmetic tasks. We hypothesize the latter, knowing that numbers are represented by their Fourier features in the token embeddings after pre-training. Consequently, this Fourier representation should be leveraged in a variety of number-related tasks. To validate this hypothesis, we perform a Fourier analysis on the GPT-2-XL model fine-tuned for the multiplication task.

Considering a maximum number of 520 for multiplication would result in an insufficient dataset size. Therefore, we set the maximum allowable product to 10000.

For each pair of numbers where the product does not exceed this limit, we used a distinct phrasing for each pair of numbers, selecting one template from five available templates. This ensures that every unique pair of numbers between 0 and 260 is presented with a consistent phrasing from these templates. We have fixed that typo in the revised version. The different phrasings used include: "What is the product of num1 and num2?", "Find the product of num1 multiplied by num2.", "Calculate num1 times num2.", "num1 multiplied by num2 equals what?", and "Multiplication of num1 with num2." The dataset is then shuffled to ensure randomness and split into training ($80\%$), validation ($10\%$), and test ($10\%$) sets. We finetune the model for 25 epochs with a learning rate of $1e-4$. Upon convergence, the validation accuracy reaches $74.58\%$.

As the primary objective is to determine whether the Fourier features are utilized in tasks other than addition, Figure 16 displays the logits in Fourier space for each layer, as in Figure 3. It is evident that the logits are sparse in Fourier space.

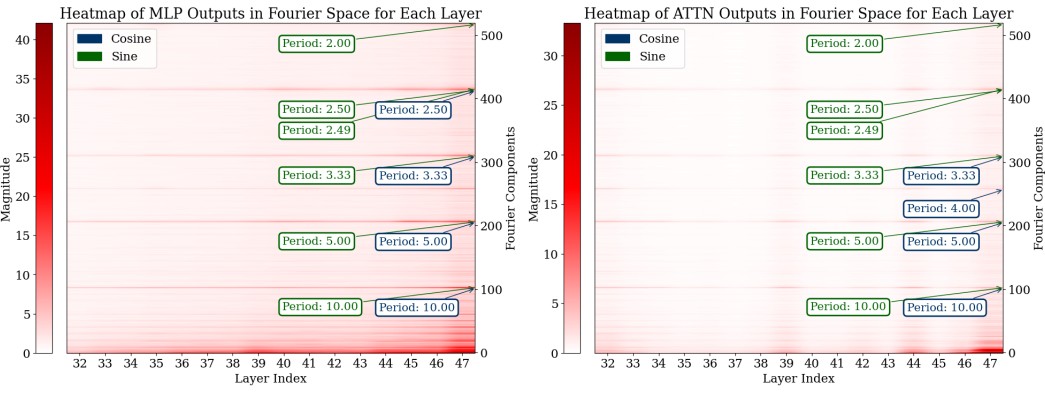

(a) Logits for MLP output in Fourier space  (b) Logits for attention output in Fourier space

Figure 16: We analyzed the logits in Fourier space for all the test data across the last 15 layers. For both the MLP and attention modules, the outlier Fourier components have periods around 2, 2.5, 3.3, 5, and 10.

## C.3 Same Results for other format

To demonstrate that our observations are not confined to a specific description of the mathematical problem, we conducted experiments on another format of addition problem and obtained consistent results. From Figure 17, we can see that there are also periodic structures in the intermediate logits.

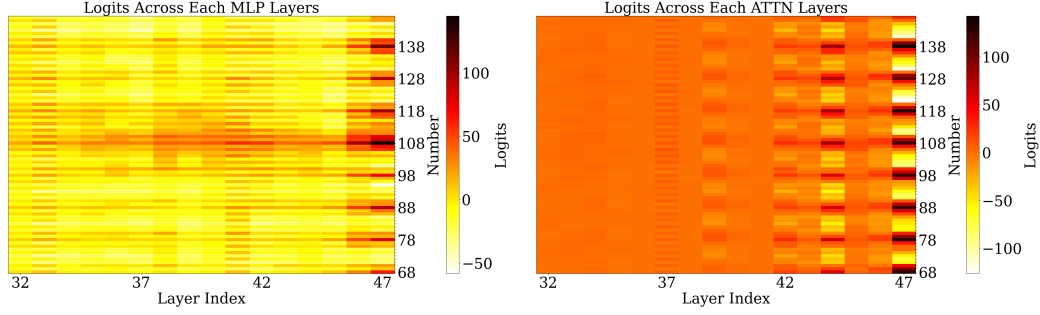

(a) MLP output logits for the last 15 layers    (b) Attention output logits for the last 15 layers

Figure 17: Heatmap of the logits across different layers. The y-axis represents the subset of the number space around the correct prediction, while the x-axis represents the layer index.

From Figure 18, we can also see the Fourier features for the MLP and attention output. These two experiments validate that our observations are not confined to a specific format of the addition problems.

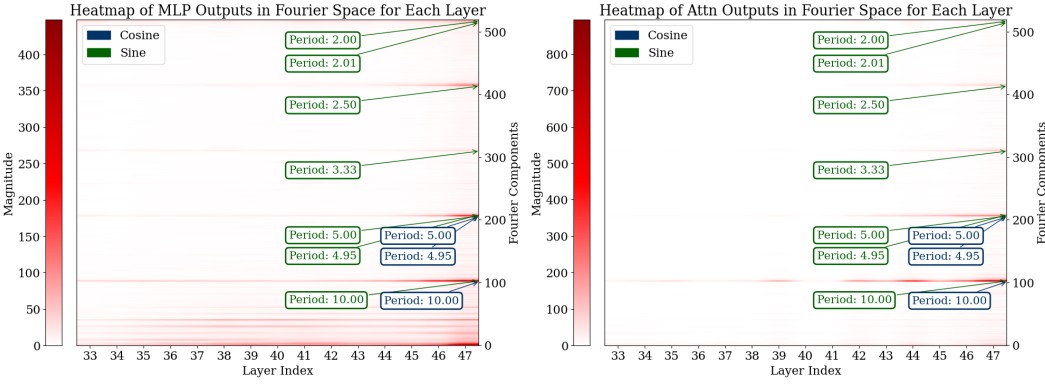

(a) Logits for MLP output in Fourier space    (b) Logits for attention output in Fourier space

Figure 18: We analyzed the logits in Fourier space for all the test data across the last 15 layers. For both the MLP and attention modules, the outlier Fourier components have periods around 2, 2.5, 5, and 10. (a) The MLP exhibits some outlier low-frequency Fourier components. (b) The attention module does not exhibit any outlier low-frequency Fourier components, but it has stronger high-frequency components.

## C.4 Fourier Features in Other Pre-trained LM

Using the Fourier analysis framework proposed in Section 3.2, we demonstrate that for GPT-J, the outputs of MLP and attention modules exhibit approximate sparsity in Fourier space across the last 15 layers (Figure 19)

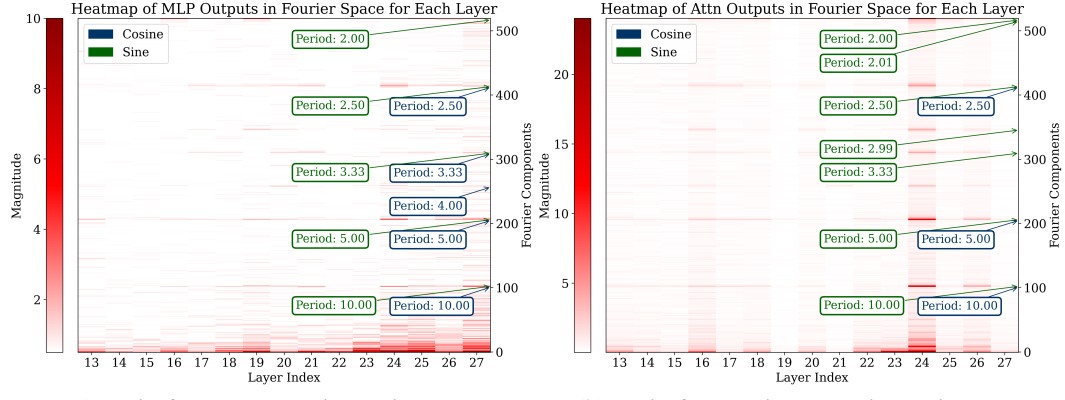

(a) Logits for MLP output in Fourier space      (b) Logits for attention output in Fourier space

Figure 19: For GPT-J (4-shot), we analyzed the logits in Fourier space for all the test data across the last 15 layers. For both the MLP and attention modules, the outlier Fourier components have periods around 2, 2.5, 5, and 10.

## D    Supporting Evidence For the Fourier Features

We selected the layers that clearly show the periodic pattern in Figure 1b and Figure 1c and plot their logits in Figure 20.

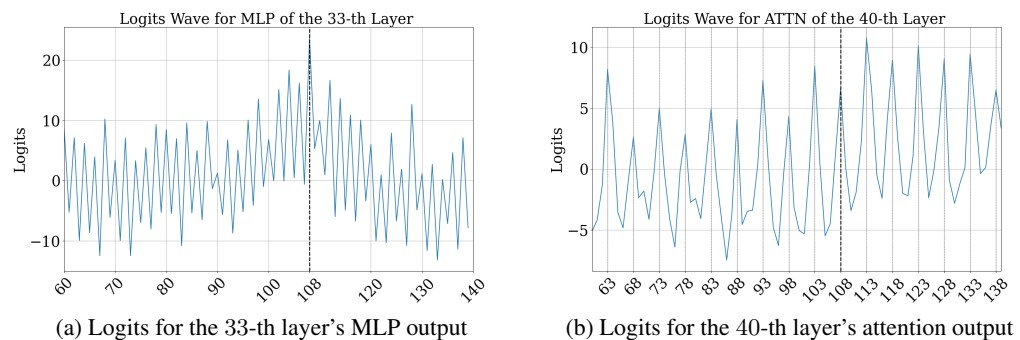

(a) Logits for the 33-th layer's MLP output      (b) Logits for the 40-th layer's attention output

Figure 20: The x-axis represents the number space, and the y-axis represents the logits value. (a) The logits wave for the MLP output of the $33^{\text{rd}}$ layer, $\mathcal{L}_{\text{MLP}}^{(33)}$. The MLP favors even numbers. The MLP module favors the answer to $15 + 93 \bmod 2$. (b) The logits wave for the attention output of the $40^{\text{th}}$ layer, $\mathcal{L}_{\text{Attn}}^{(40)}$. The attention module favors the answer to $15 + 93 \bmod 10$ and $15 + 93 \bmod 5$.

Figure 21 illustrates that the errors resulting from the ablation study (Section 3.3) correspond with our theoretical insights. Removing low-frequency parts from the MLP results in errors such as off-by 10, 50, and 100. Without these low-frequency components, the MLP is unable to accurately approximate, although it still correctly predicts the unit digit. In contrast, removing high-frequency components from the attention modules results in smaller errors, all less than 6 in magnitude. These findings support our statement that low-frequency components are essential for accurate approximation, whereas high-frequency components are key for precise classification tasks. Consequently, the primary function of MLP modules is to approximate numerical magnitudes using low-frequency components, and the essential function of attention modules is to facilitate precise classification by identifying the correct unit digit.

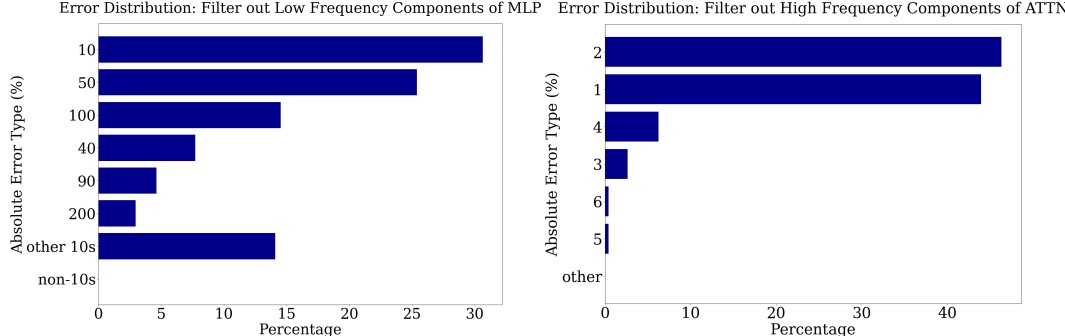

(a) Filtering out low-frequency components of MLP   (b) Filtering out high-frequency components of attention

Figure 21: (a) Across all the test data, the difference between predictions and labels are all the multiple of 10. (b) Across all the test data, the differences between predictions and labels are all below 6.

# E   More Experiments on GPT-2-XL Trained from Scratch

Following the methodology proposed in Section 3, we plotted the logits of the MLP and attention modules for each layer, as shown in Figure 22. The prediction is solely determined by the 40-th layer MLP. Unlike Figure 3, there is no observable periodic structure across all layers.

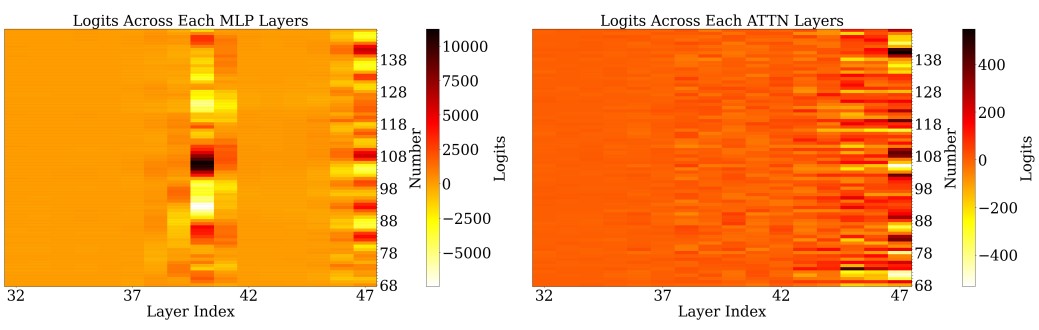

(a) MLP output logits for the last 15 layers   (b) Attention output logits for the last 15 layers

Figure 22: Heatmap of the logits across different layers. The y-axis represents the subset of the number space around the correct prediction, while the x-axis represents the layer index. The final prediction is solely decided by the 40-th layer MLP.

For the model trained from scratch on the created addition dataset, all of the predictions on the test dataset deviate from the correct answer within 2 as shown in Figure 23.

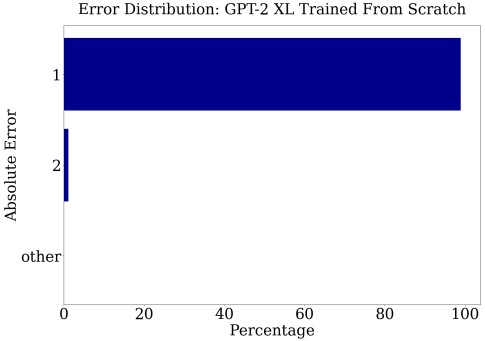

Figure 23: Error distribution for GPT-2-XL trained from scratch.

# F  Details of Experimental Settings

**Fine-tuned GPT-2-XL** We finetune GPT-2-XL on the "language-math-dataset" with 50 epochs and a batch size of 16. The dataset consists of $27,400$ training samples, $3,420$ validation samples, and $3,420$ test samples. We use the AdamW optimizer, scheduling the learning rate linearly from $1 \times 10^{-5}$ to 0 without warmup.

**Train GPT-2-XL from scratch** We train GPT-2-XL on the "language-math-dataset" from scratch with 500 epochs and a batch size of 16. The dataset consists of $27,400$ training samples, $3,420$ validation samples, and $3,420$ test samples. We use the AdamW optimizer, scheduling the learning rate linearly from $1 \times 10^{-4}$ to 0 without warmup.

**Train GPT-2 from scratch** For both with pre-trained token embedding and without token embedding, we train GPT-2 on the "language-math-dataset" with 700 epochs and a batch size of 16. The dataset consists of $27,400$ training samples, $3,420$ validation samples, and $3,420$ test samples. We use the AdamW optimizer, scheduling the learning rate linearly from $5 \times 10^{-5}$ to 0 without warmup. In Figure 7b, we train the model with five different seeds and plot the mean and deviation for them.

**Create the addition dataset in main content** We consider numbers in base 10 up to a maximum value of 260. For each pair of numbers between 0 and 260, we used a distinct phrasing for each pair of numbers, selecting one template from five available templates. This ensures that every unique pair of numbers between 0 and 260 is presented with a consistent phrasing from these templates. We have fixed that typo in the revised version. The different phrasings used are: "Total of num1 and num2.", "Add together num1 and num2.", "Calculate num1 + num2.", "What is the sum of num1 and num2?", and "Put together num1 and num2.". The dataset is shuffled to ensure randomness and then split into training ($80\%$), validation ($10\%$), and test ($10\%$) sets.

**Create the addition dataset in Appendix C.3 with different format** We consider numbers in base 10 up to a maximum value of 260. We generate all possible pairs of numbers within this range using combinations with replacement. For each pair, we convert the numbers to the specified base and create questions formatted as "num1,num2+" with their corresponding answers. The dataset is then split into training ($80\%$), validation ($10\%$), and test ($10\%$) sets.

**Experiments Compute Resources** All experiments involving fine-tuning and training from scratch in this paper were conducted on one NVIDIA A6000 GPU with 48GB of video memory. The fine-tuning process required less than 10 hours, while training from scratch took less than 3 days. Other experiments, such as those involving Logit Lens, were completed in less than 1 hour.

**Licenses for Existing Assets & Open Access to Data and Code.** For the following models, we use the checkpoints provided by Huggingface. For all the trained models, we use default hyperparameters during all the training but with different random seeds.

- GPT-2-XL: https://huggingface.co/openai-community/gpt2-xl, Modified MIT License
- GPT-2: https://huggingface.co/openai-community/gpt2, Modified MIT License
- GPT-J: https://huggingface.co/EleutherAI/gpt-j-6b, Apache-2.0 License
- Phi2: https://huggingface.co/microsoft/phi-2, MIT License

- GPT-3.5 and GPT-4: https://chatgpt.com/ or https://openai.com/index/openai-api/
- PaLM-2 https://ai.google/discover/palm2/

# G Impact Statement

Our work aims to understand the potential of large language models in solving arithmetic tasks. Our paper is an interpretability paper and thus we foresee no immediate negative ethical impact. We believe improved understanding and enhancement of LLMs can lead to more robust AI systems that are capable of performing complex tasks more reliably. This can benefit areas such as automated data analysis, financial forecasting, and more.

