# OpenReview forum: "Pre-trained Large Language Models Use Fourier Features to Compute Addition"
_NeurIPS.cc/2024/Conference — NeurIPS 2024 poster_

### Official Review · Reviewer_mHEL · 2024-06-26

**Soundness:** 2
**Presentation:** 3
**Contribution:** 2
**Rating:** 5
**Confidence:** 3

**Summary:**

The authors utilize discrete Fourier transform to determine which Fourier components play the most important role in computing the addition of relatively small numbers (< 520) in Large Language Models, such as the GPT-2 family, and, _likely_, Phi-2, GPT-J, and others. They found out that MLP modules of GPT-2 use mostly low-frequency Fourier components to approximate the magnitude of the answer. In contrast, MHA modules utilize high-frequency components to approximate the last digit of the answer. Moreover, the model converges to the correct answer layer by layer.

The paper shows that this mechanism appears after pretraining the model; the weights of the embedding layer play a special role in it.

**Strengths:**

The paper provides interesting insights into the addition mechanism inside of Large Language models:

- It contains original research about the role of Fourier components in small numbers addition inside of GPT-2 family models;
- it provides some observations that indicate that the same mechanism may appear in Phi-2, GPT-J, and bigger models;
- it demonstrates how the roles of MHA and MLP complement each other in this mechanism;
- it shows the connection between this mechanism and the pretraining of the model, especially the embedding layer.

**Weaknesses:**

- Most of the experiments were devoted to the models from the GPT-2 family; very few experiments have been carried out on other models, so any arguments about other models are weaker;
- Only small (< 520) numbers were considered;
- The experimental setup is questionable (see "Questions" part of the review);
- The paper is not easy to follow. Some figures are confusing, and there are many typos (see "Questions" part of the review).

**Questions:**

Questions:

- If I understood correctly, Table 1 contains the ablation experiments only for GPT-2-XL. Why didn't you repeat the experiments from Table 1 for other models? E.g. for Phi2, GPT-2-base, etc.
- How does the mechanism change when you add the bigger numbers, encoded in several tokens instead of one token?
- You wrote in 91 lines:
> ℓ ∈ [L].

Does it mean, that you use the outputs of all layers, preceding layer number L, for prediction?
- You wrote at 594 lines:
> We consider numbers in base 10 up to a maximum value 260. For each pair of numbers between 0 and 260, we generate various phrasings of addition questions and their corresponding answers. The different phrasings used are: “Total of num1 and num2.”, “Add together num1 and num2.”, “Calculate num1 + num2.”, “What is the sum of num1 and num2?”, and “Put together num1 and num2.”. The dataset is shuffled to ensure randomness and then split into training (80%), validation (10%), and test (10%) sets

Does it mean that train, valid and test could contain the addition questions for the same numbers, but in different phrasing? I.e., could the train contain the phrase “Total of 11 and 12.”, while the test contains “Put together 11 and 12.”?

---

Suggestions:

- GPT-2 is a model with absolute position embeddings. What about other types of position embeddings? (see also my question/suggestion about ablation experiments on Phi2 above). It would be interesting to research the connection of the addition mechanism with the position embedding type. See the paper "**Positional Description Matters for Transformers Arithmetic**" by Ruoqi Shen, Sébastien Bubeck, Ronen Eldan, Yin Tat Lee, Yuanzhi Li, and Yi Zhang.
- It would also be interesting to research the connection between the Fourier addition mechanism and outlier dimensions (see, for example, the paper "**Outlier Dimensions Encode Task-Specific Knowledge**" by William Rudman, Catherine Chen, Carsten Eickhoff, and many other papers about Outlier Dimensions in transformers).

---

Typos and presentation issues:

- Figures 1 (b) and (c) are very confusing because the word "Number" is closer to the color bar, than the word "Logits". So, at first, I didn't understand, that the color means the **Logit value**. As a result, I was confused about these diagrams, and it took time for me to sort things out.
- You wrote in line 165:
> We show in Appendix A that this has a simple closed-form solution involving a linear projection.

I didn't find this in Appendix A, can you please, elaborate, on what you meant?
- You wrote in the description of Figure 13:
> Visualization of the Fourier component whose period is 520 analysis for the final logits.

Can you please, elaborate, on what did you mean?
- You wrote in line 204:
> D is the size of the token embeddings, denote the token embedding for numbers

Can you please, elaborate, on what did you mean?

- What does Fig.5(a) show? Are these some average number embeddings?
- Why does Fig.6 show the legend with the different colors of cos/sin, while only red bars are visible? Maybe the bars are too narrow, so their colors mix together?

**Limitations:**

There are two small "Limitation" and "Impact" sections: Appendix G Limitations, and Appendix H Impact Statement.

The authors addressed the limitations correctly overall. However, I would add that most of the experiments were devoted to the models from the GPT-2 family; very few experiments have been carried out on other models, so any arguments about other models are weaker. It is also a big limitation of this work.

---

> ### Author Rebuttal · Authors · 2024-08-06
>
> Thank you for the supportive feedback and comments.
>
>  ***Q1: Why not repeat the experiments from Table 1 for other models.***
>
> The goal of this paper is to understand how pre-trained LLMs solve addition tasks. We conduct most of the experiments, such as Figures 1-5, on GPT-2-XL to provide a deep and comprehensive understanding of its mechanisms in solving these tasks. We focus on GPT-2-XL because it offers a balance between model complexity and interpretability, allowing us to draw meaningful insights.
> To address the generalizability of our findings, we provide evidence for the existence of Fourier features in other pre-trained LLMs and tasks. Specifically, our Fourier analysis of the pre-trained embeddings for different models (as shown in Figure 14) reveals a consistent sparsity in Fourier space, even without fine-tuning. Additionally, the intermediate logits (Figure 8) and final predictions (Section 4.3) further demonstrate this sparsity across models.
>
> These results suggest that the observed phenomena are not unique to GPT-2-XL but are likely intrinsic to the architecture of pre-trained LLMs in general. Therefore, while repeating all experiments on every other pre-trained LLM would be ideal, we believe the provided evidence sufficiently supports our claims across different models. This approach balances thoroughness with practicality, ensuring that our main findings are robust and broadly applicable without unnecessary redundancy.
>
>
> ***Q2: In 91 lines: $\ell \in [L]$. Does it mean that you use the outputs of all layers, preceding layer number L, for prediction?***
>
> No, $[L]$ stands for $\{1,2,3,4….L\}$. Hence, $h^{(\ell)}$ where $\ell \in [L]$,  refers to one specific hidden state on the residual stream at layer $\ell$.
>
> ***Q3: GPT-2 is a model with absolute position embeddings. It would be interesting to research the connection of the addition mechanism with the position embedding type. See the paper "Positional Description Matters for Transformers Arithmetic"***
>
> Thank you for your comments. In our paper, each number is treated as one token, corresponding to one token embedding (as mentioned in line 58). Therefore, the type of positional embedding does not change our results and observations.
> Regarding the paper "Positional Description Matters for Transformers Arithmetic," it's important to note that in the footnote of page 3, they mention: “In this paper, for every dataset used, a space is inserted before each digit. This ensures the tokenizer tokenizes each digit as an individual token.” This approach allows them to leverage the position of each digit to solve arithmetic tasks, which is not a common way to tokenize numbers in the GPT-2 model in real-world scenarios.
>
>
> ***Q4: It would also be interesting to research the connection between the Fourier addition mechanism and outlier dimensions (paper "Outlier Dimensions Encode Task-Specific Knowledge").***
>
> Thank you for your comments. Outlier dimensions are the dimensions in the embedding space that exhibit large variance. In our work, we demonstrate that there are outlier components in the Fourier space that play a significant role in arithmetic tasks. We emphasize that the outlier are in two different space for these two cases.
>
> ***Q5: Figures 1 (b) and (c) are very confusing***
>
> Thanks for your feedback. We have revised Figures 1(b) and 1(c) to improve clarity by adjusting the placement of labels and color bars.
>
> ---
> ***For Q6 - Q9, Reviewer mHEL requested further elaboration on the processes, figures, and concepts discussed in the paper. Due to space constraints, we have addressed these details in the official comment.***
>
>
> ---
>
> ***Q10: Why does Fig.6 show the legend with the different colors of cos/sin, while only red bars are visible?***
>
> Thanks for noticing that. We used the same plotting method in Figure 3 and Figure 6. In Figure 3, we use the legend with different colors to show whether the outlier component is sine or cosine. However, for Figure 6, there are no outlier Fourier components and we do not need that legend. We have removed the legend in Figure 6 in the revised version of our paper.
>
>
> ***Q11: The authors addressed the limitations correctly overall. However, I would add that most of the experiments were devoted to the models from the GPT-2 family; very few experiments have been carried out on other models, so any arguments about other models are weaker. It is also a big limitation of this work.***
>
> We acknowledge that this paper primarily focuses on the GPT-2 family. However, we believe that this already constitutes a significant step forward, highlighting our contribution rather than a limitation. As discussed in the related work section, prior studies [1,2,3,4] mainly concentrated on training shallow networks from scratch and analyzing their performance in modular addition. To the best of our knowledge, this is the first work analyzing how pre-trained LLMs solve addition.
> We delve deeply into understanding the mechanisms by which GPT-2 leverages Fourier features. We then provide evidence of the existence of these Fourier features in other closed-source and open-source pre-trained LLMs. We demonstrate similar Fourier features in both pre-trained number embeddings and the intermediate hidden states. Additionally, these outlier Fourier components in other models exhibit almost the same periods as those in GPT-2. Based on these findings, we conclude that they leverage Fourier features to solve addition.
>
> [1] Morwani, Depen, et al. "Feature emergence via margin maximization: case studies in algebraic tasks."
>
> [2]Nanda, Neel, et al. "Progress measures for grokking via mechanistic interpretability."
>
> [3]Zhong, Ziqian, et al. "The clock and the pizza: Two stories in mechanistic explanation of neural networks."
>
> [4]Gu, Jiuxiang, et al. "Fourier circuits in neural networks: Unlocking the potential of large language models in mathematical reasoning and modular arithmetic.

---

> ### Author Response · Authors · 2024-08-06
> **Detailed Elaborations: Part 1**
>
> ***Q6: In line 165: We show in Appendix A that this has a simple closed-form solution involving a linear projection. I didn't find this in Appendix A, can you please, elaborate, on what you meant?***
>
> Thanks for your comments. In Definition A.6 (line 476 in Appendix A), we formally define the low/high-pass filter used in Section 3.3 as an optimization problem. We provide the closed-form solution to this problem in Remark A.5 (line 483 in Appendix A). To clarify this closed-form solution, let's walk through the derivation behind it step by step.
>
> **Problem Statement**
>
> We aim to filter the vector $x \in \mathbb{R}^D$ using a high-pass or low-pass filter, defined by the following optimization problem:
>
> $
> \min_y \| x - y \|_2^2 \quad \text{subject to} \quad B F W_U y = 0
> $
>
> Here, $B$ is a diagonal binary matrix that selects which frequency components to retain, $F$ denotes the Fourier basis, and $W_U$ is the output embedding matrix.
>
> **Intuition Behind the Solution**
>
> The objective is to find the vector $y$ that is as close as possible to $x$ while satisfying the constraint $B F W_U y = 0$. This constraint ensures that the filtered output $y$ lies in the null space of $B F W_U$.
>
> **Null Space Projection**
>
> To solve this optimization problem, we project $x$ onto the null space of $B F W_U$. The null space $N(B F W_U)$ consists of all vectors $v$ such that $B F W_U v = 0$. The projection of any vector $x$ onto this null space gives us the closest vector in the null space to $x$.
>
> **Deriving the Projection**
>
> The objective function $\| x - y \|_2^2$ measures the Euclidean distance between $x$ and $y$. Minimizing this function ensures that $y$ is as close to $x$ as possible. The constraint $B F W_U y = 0$ ensures that $y$ lies in the null space of $B F W_U$. To satisfy the constraint while minimizing the distance, we need to project $x$ onto the null space of $B F W_U$. The projection operator onto the null space $N(B F W_U)$ can be represented by the matrix $N(B F W_U) N(B F W_U)^\top$.
>
> **Closed-form Solution**
>
> The closed-form solution to the optimization problem is obtained by projecting $x$ onto the null space of $B F W_U$:
>
> $
> y = P_{N(B F W_U)} x = N(B F W_U) N(B F W_U)^\top x
> $
>
> Here, $N(B F W_U)$ is the basis for the null space of $B F W_U$, and the matrix $N(B F W_U) N(B F W_U)^\top$ represents the projection operator.
>
>
>
>
>
> ***Q7: You wrote in the description of Figure 13: Visualization of the Fourier component whose period is 520 analysis for the final logits. Can you please, elaborate, on what did you mean?***
>
> Thank you for your comments. Let us emphasize the significance of Figure 13 and its implications. In our paper, we separate the frequency components into low-frequency and high-frequency components. We state that MLP layers primarily approximate the magnitude of the answer using low-frequency features, while attention layers primarily perform modular addition. One may ask: why can't we just use low-frequency components for classification? For example, ‘a+b mod 520’ will give the correct answers in our dataset. Figures 12, 13, and lines 515-526 in the appendix address this question.
>
> When the model computes ‘a+b mod 2’, it is equivalent to a binary classification problem, which is simpler than ‘a+b mod 520’ (a 520-class classification). Figure 12b shows that the model accurately places the peak of the wave at the answer for ‘a+b mod 2’, ‘mod 5’, and ‘mod 10’. However, for large-period components, Figure 12a shows that the model only approximates the answer for ‘a+b mod p’, where p is the period of the components.
>
> As we need to show all the large-period components in Figure 12a, we cannot display the entire number range. Therefore, the reader cannot tell whether the wave with a period of 520 correctly places its peak at the correct answer. Hence, we include Figure 13, which shows the period-520 wave over the entire number range. Figure 13 clearly demonstrates that the period-520 wave fails to place its peak on the correct answer and only approximates it.
>
> As discussed in lines 129-136, the final logit is the accumulation result from the output of all the layers and is sparse in Fourier space. Figures 12 and 13 imply that the accumulation of low-frequency (large-period) components across all layers cannot make the correct prediction for ‘a+b mod p’ for large periods ‘p’. This provides evidence for why high-frequency (small-period) components are necessary for the model to make accurate predictions.

---

> ### Author Response · Authors · 2024-08-06
> **Detailed Elaborations: Part 2**
>
> ***Q8: You wrote in line 204: D is the size of the token embeddings, denote the token embedding for numbers Can you please, elaborate, on what did you mean?***
>
> In line 204, we wrote ‘Let $W_E \in \mathbb{R}^{p \times D}$, where $p = 521$ and $D$ is the size of the token embeddings, denote the token embedding for numbers.’ By this, we mean that $W_E$ denotes the token embedding matrix for numbers. Normally the token embedding has a shape (vocabulary size, token embedding size). However, since we only consider the integer in [0,520], the vocabulary size becomes the size of the number space, and we have $p=521$.
>
> ***Q9: What does Fig.5(a) show? Are these some average number embeddings?***
>
> Fig. 5(a) is not an average number embedding. Lines 203-216 explain Figure 5(a). Let us elaborate on the process of creating Figure 5(a) step by step to help you gain a better understanding.
> 1. For each pre-trained LLM, there is a pre-trained token embedding matrix. Since we only consider the embeddings for operands in the range [0, 520], we select the 521 rows corresponding to these 521 numbers from the token embedding matrix. We define this selected token embedding as the number embedding $E$ for simplicity. This number embedding matrix’s shape is (size of embedding dimension, 521).
> 2. Next, we apply the Discrete Fourier Transform to this number embedding matrix. We multiply this number embedding matrix by a Fourier basis, whose shape is (521, 521), resulting in the number embedding matrix in Fourier space. The shape of the number embedding matrix in Fourier space is (size of embedding dimension, 521). Each entry represents the magnitude of each Fourier component distributed to each embedding dimension.
> 3. Then, we take the L2 norm along the embedding dimension and obtain a vector ‘v’ with size 521. Each entry reflects the magnitude of each Fourier component distributed over the embedding dimension. Intuitively, each entry in this vector shows how important each Fourier component is in constituting the number embedding.
> 4. We ignore the constant term in that vector and plot the remaining values in Figure 5(a). According to the construction of the Fourier basis (Definition A.3), v[i] represents the sine wave when ‘i’ is even, and the cosine wave when ‘i’ is odd. Hence, we obtain the plot shown in Figure 5(a).
> 5. Figure 5(a) demonstrates that the number embedding has outlier components with periods of 2, 2.5, 5, and 10. The insight here is that these outlier Fourier components represent the numbers in the pre-trained GPT2-XL model. Compared with Figure 7(a), we show that using Fourier features to embed numbers is a learned behavior during pre-training. Additionally, we provide further evidence in Figure 14 to show that pre-trained LLMs tend to use Fourier features to embed numbers.
>
> ***If you need any further elaboration, please let us know.***

---

> ### Author Response · Authors · 2024-08-12
>
> As the discussion period approaches the deadline, we would like to follow up to see if our response and elaboration above has addressed your questions. Again, thanks for your time and effort. Looking forward to hear from you.

---

### Official Review · Reviewer_wYe3 · 2024-07-11

**Soundness:** 3
**Presentation:** 3
**Contribution:** 3
**Rating:** 6
**Confidence:** 4

**Summary:**

This paper focuses on understanding the mechanisms the LLMs employ to carry out mathematical operations, in particular sum of two numbers.  It demonstrates: a) Models utilize Fourier features, with different parts of the model utilizing different frequency ranges — attention mechanism uses high frequency components while MLP primarily uses low frequencies.  This is further corroborated by conducting filtering studies.  b) Model embeddings and the pre-training process play a critical role in learning these Fourier features.  Models that are not pre-trained do not seem to be able to induce Fourier features and have lower accuracy.  However, such models when seeded with only embeddings from pre-trained models do learn the Fourier features.

**Strengths:**

* Well written paper on understanding basic mechanism of how LLMs carry out the specific task of addition of small numbers.
* Interesting findings on pre-training vs task specific training from scratch, demonstrating criticality of model pre-training.

**Weaknesses:**

* Study is limited to addition and that too of numbers where the final sum can be represented by a single token, leaving a gap in understanding of how addition of larger numbers spanning multiple tokens is achieved.
* With use of tools / functions in conjunction with the LLMs, mathematical operations are often carried out using a calculator or similar tool.   Thus attempting to improve LLMs' arithmetic abilities may not be very fruitful.  However, do the discoveries made in the paper apply only to number processing by LLMs or can they have broader implications, say on logical reasoning?

**Questions:**

Lines 154-156 : “We also illustrated that in one example, the high-frequency components primarily approximate the magnitude, while the low-frequency components are crucial for modular addition tasks, as depicted in Figure 4.“  Is it the other way around?

**Limitations:**

Limitations sufficiently addressed

---

> ### Author Rebuttal · Authors · 2024-08-06
>
> Thank you for the supportive feedback and thoughtful comments.
>
> ***Q1: With the use of tools / functions in conjunction with the LLMs, mathematical operations are often carried out using a calculator or similar tool. Thus attempting to improve LLMs' arithmetic abilities may not be very fruitful. However, do the discoveries made in the paper apply only to number processing by LLMs or can they have broader implications, say on logical reasoning?***
>
> Thank you for your feedback. We agree that using a calculator or similar tool can solve arithmetic problems easily. However, we believe that without understanding numbers and performing basic arithmetic, a model cannot fully grasp more complex concepts in physics or mathematics like humans do. With billions of parameters, LLMs should be capable of easily solving number-related tasks such as arithmetic problems and time-series prediction. Thus, we believe improving LLMs' arithmetic abilities is important.
>
> Additionally, we emphasize that our observations have broader implications beyond arithmetic tasks. One key finding is that number embeddings after pre-training are sparse in the Fourier space for many pre-trained LLMs without fine-tuning (Figure 14). This suggests that the model inherently uses Fourier features to represent numbers and solve number-related tasks. This insight can inspire future research in several ways:
> - Extending this number embedding strategy to handle larger numbers.
> - Adding regularizers to help models learn high-frequency Fourier features to enhance performance on number-related tasks.
> - Exploring the potential for these Fourier features to benefit logical reasoning and other tasks beyond arithmetic.
>
> By understanding and leveraging these Fourier features, we can improve LLMs' overall capabilities in number processing and related areas, thereby contributing to their broader applicability and effectiveness. More broader implications and future directions can be found in our response to all reviewers.
>
> ***Q2: Lines 154-156 : “We also illustrated that in one example, the high-frequency components primarily approximate the magnitude, while the low-frequency components are crucial for modular addition tasks, as depicted in Figure 4.“ Is it the other way around?***
>
> Thank you for pointing out that typo. We have revised it to: “We also illustrated that in one example, the low-frequency components primarily approximate the magnitude, while the high-frequency components are crucial for modular addition tasks, as depicted in Figure 4.”

---

### Official Review · Reviewer_XwiY · 2024-07-13

**Soundness:** 3
**Presentation:** 4
**Contribution:** 3
**Rating:** 7
**Confidence:** 4

**Summary:**

This paper analyzes how language models perform addition, showing that they use Fourier features. Most of the analyses are done on fine-tuned models. First, periodicity is shown with a logit-lens technique on different layers. Fourier analysis shows this nicely in the frequency space, and shows differences between MLP and attention components, the latter exhibiting only high-freq components. This is connected to "approximation" of the answer vs "classification" per small modulo. Ablation by removing low/high frequency supports a difference in behavior of the two components. The next section shows that models trained from scratch do not have such Fourier features, but plugging in trained number embeddings recovers this behavior. Finally some analysis shows a similar pattern in prompted models.

**Strengths:**

1. Seemingly novel observations about language models' mechanism of addition computation.
2. Clear and to-the-point analyses and visualizations.
3. Both correlational and causal experiments via ablations to raise hypotheses and test them.

**Weaknesses:**

1. Missing quantitative analysis of the superposition pattern in section 3.2. See below in questions.
2. While the main experiments in section 3 (and to a lesser extent also 4.1 and 4.2) are quite thorough and convincing, the ones in 4.3 and in some places in the appendix are less detailed and not sufficiently contrasted with the main results. See below in questions.
3. Potential problem with datasplit having leakage of expressions from training to test. See below in questions.

**Questions:**

1. Data split seems to allow cases where the same expression (say 12 + 17) appears in both training and test sets, just with different templates. When looking at generalization, especially when fine-tuning a pretrained model, it could be beneficial to make sure no such overlap happens.

2. Section 3.1 assumes that "If the models merely retrieve and recombine pieces of information learned during training, certain layers will directly map this information to predictions". I am not convinced by this. I don't see why 'memorization and recombination' cannot be implemented by the full model, with this gradual refinement process as an implementation of memorization.

3. The periodicity analyses in section 3.2 are striking and insightful. That said, one concern is quantitative analysis - I found the example in figure 4 helpful, but would appreciate a quantitative evaluation of the superposition phenomenon. Also, given the residual stream view, one could say that this is trivial. That is, the final logits are always the sum of earlier ones. So, it would be useful to explain the significance of this finding and whether there is anything special in this case of addition compared to the general behavior of the model. Even better, some control test in a quantitative evaluation could make this point more compelling.

4. Section 3.3 is excellent - it shows which components are causally important for which roles. The experiments are well planned and convincing. One comment I have is on line 183, which says "the approximation tasks are primarily performed by the MLP modules alone". If I read Table 1 correctly, then, given that removing low-freq from both Attn and MLP leads to an even greater reduction in accuracy, we can conclude that both MLP and Attn are involved in approximation. This is essentially the same argument made in the paper just a few lines above.

5. Since the analysis in section 3 is done on fine-tuned model, I don't think we can conclude that "The previous section shows that pre-trained LLMs leverage Fourier features to solve the addition problem" (line 195). The rest of the section in fact gives evidence to this, but the phrasing at this point should be revised.

6. Figure 3 is very nice but other figures aren't as clean, specifically figure 8 (4-shot) and figure 15 (multiplication), figure 17 (other format), and figure 18 (GPT-J). I think the discussion of each of these cases was a bit too optimistic and naive, and the comparison to the main analyzed case should be made more carefully, and differences highlighted.

7. The related work section is well done. Clarifying the difference from [28] early on in the paper could be useful for the reader. And, one can also mention this paper on arithmetic mechanisms in the related work: Stolfo et al., A Mechanistic Interpretation of Arithmetic Reasoning in Language Models using Causal Mediation Analysis.

**Limitations:**

Sufficient discussion.

---

> ### Author Rebuttal · Authors · 2024-08-06
>
> Thank you for the supportive feedback and thoughtful comments.
>
> ***Q1: Why 'memorization and recombination' cannot be implemented by the full model, with this gradual refinement process as an implementation of memorization.***
>
> Thank you for your feedback. Our assertion is based on prior research and specific empirical observations detailed in Section 3.1. [1,2] indicate that only a few late layers integrate information for prediction in fact-retrieval tasks. However, our findings in the addition task (Figure 1a) demonstrate that the model utilizes over 20 layers to compute the result, suggesting a more complex process than mere retrieval and recombination. This extensive use of layers aligns with the model performing actual computation, progressively refining the output across multiple layers, contrary to what would be necessary if it were merely memorizing and recombining pre-learned facts.
>
> Furthermore, in Figure 1a, we can see the results gradually refining from far from the correct result to accurate as processing progresses through the layers, which clearly show that the model is performing computation.
>
> [1] Meng, Kevin, et al. "Locating and editing factual associations in GPT."
>
> [2] Merullo, Jack, Carsten Eickhoff, and Ellie Pavlick. "Language models implement simple word2vec-style vector arithmetic."
>
> ***Q2: Requests quantitative evaluation of the superposition phenomenon (Figure 4) in periodicity analyses and questions the triviality of findings given the model's architecture.***
>
> Thanks for your insightful comments. We emphasize that Figure 4 shows different dimensions in the Fourier space for the final logits. Hence, It is not the sum of the logits of earlier ones. The goal of that experiment (Figure 4) is to provide readers with an intuition of how the final logits are expressed with only a few Fourier components and why sparse Fourier components are sufficient to make the prediction.
>
> In Figures 2 and 3, we perform quantitative analysis in Fourier space to show that each layer’s contribution to the residual stream is sparse in Fourier space. This Fourier space analysis may not intuitively explain how these outlier components form the final logits. Therefore, we believe that by showing the final logits with only a few Fourier components in number space for this specific example, readers can better understand how these outlier Fourier components contribute to the prediction.
>
> ***Q3: Critique of the statement that approximation tasks are primarily performed by MLP modules, suggesting involvement of both attention and MLP modules.***
>
> Thank you for your comments. Regarding the statement on line 183, "the approximation tasks are primarily performed by the MLP modules alone," we would like to emphasize our findings as follows:
> In Table 1, we observe that removing low-frequency components from the attention output alone does not lead to a significant reduction in accuracy. However, when we remove low-frequency components from both the attention and MLP modules, there is a larger reduction in accuracy compared to removing them from the MLP modules alone. This suggests that while both attention and MLP modules are involved in approximation, the MLP modules play a more critical role.
> Furthermore, the minimal impact on accuracy from removing low-frequency components from the attention output indicates that the MLP modules are capable of performing the approximation tasks independently. During prediction, the low-frequency components of the attention output do not contribute significantly, as the MLP modules have the ability to handle the approximation effectively on their own.
> Therefore, we conclude that “the approximation tasks are primarily performed by the MLP modules alone,” as the MLP modules are sufficient to achieve the desired approximation even in the absence of low-frequency components from the attention output.
>
> ***Q4: Suggestion to revise phrasing regarding the use of Fourier features in pre-trained LLMs due to the analysis being on a fine-tuned model.***
>
> Thanks to the reviewer for pointing that out. You are correct. That is one of our observations in Section 4. We have revised that sentence to “The previous section shows that pre-trained LLMs leverage Fourier features to solve the addition problem after fine-tuning”.
>
> ***Q5: Figure 8,15, 17,18 are not as clean as Figure 3***
>
> Thank you for your detailed feedback. We have updated the appendix in our paper to incorporate more discussion about the figures you mentioned, which will be reflected in the revised manuscript. We put the full version in the official comment. Here is a summary of the updated discussion.
>
> In the analysis of various figures, Figure 17 highlights the effective learning of the 'mod 10' task by the last few layers, showing more prominent outlier frequency components compared to Figure 3. Figures 8 and 18 exhibit similar outlier components to Figure 3 but with less clarity, indicating that without fine-tuning, models like 4-shot Phi2 and GPT-J (with respective accuracies of 55.44% and 72.21%) struggle to fully utilize Fourier features for arithmetic tasks. This observation suggests potential improvements in model performance through enhanced leveraging of these features. Meanwhile, Figure 15 reflects the complexity models face with multiplication tasks, where despite data expansion and fine-tuning, accuracies remain relatively lower (74.58%) due to inadequate utilization of Fourier features, as also supported by recent studies from McLeish et al. (2024) and Dziri et al. (2024).
>
> ***Q6: Clarify the difference from Nanda et al. (2023) and add Stolfo et al. (2023) to related work.***
>
> Thank you for your supportive comments. We have added more discussion for Nanda et al. (2023) and Stolfo et al. (2023) to the related work section, which will be reflected in the revised manuscript.  Due to space limits, we put the fully revised version in the official comment.

---

> > ### Comment · Reviewer_XwiY · 2024-08-12
> > **Thanks for the response; missing one important question**
> >
> > Thank you for your response.
> >
> > I'd love to see a response about my concern regarding the data split in my question 1.
> >
> > 1. Concerning memorization vs computation/recombination, I guess it's partly philosophical and hard to prove, but I don't see "direct output" as a requirement for memorization. I think memorization can be implemented by a gradual computation process (even if in other, retrieval-oriented tasks, it's more direct).
> >
> > 2. Thanks for the clarification about figure 4. Still, figures 2+3 don't quantify the superposition phenomenon, do they? Would there be a way to quantify this specifically?
> >
> > 3. I still think that the statement "primarily performed by the MLP modules alone," is inaccurate, since attention modules are involved, as you also state. There may be some redundancy, but the above statement seems misleading.
> >
> > 4. Thanks for a more careful discussion of other figures besides 3 and look forward to seeing it in the paper's next revision. This statement is excellent:
> > " we believe that without fine-tuning, the model does not fully leverage Fourier features to solve the addition problem. Even though the pre-trained model has learned to represent the numbers in Fourier space, it fails to fully understand the question phrasing and apply the Fourier method to solve it. We believe that better leveraging the Fourier features to improve the models’ performance on arithmetic tasks can be an interesting direction."
> >
> > I believe something along these lines should appear in the main body of the paper, to accurately represent the contribution of the present work.
> >
> > 5. Your clarifications about differences from refs [1], [2] are helpful; thank you.

---

> > > ### Author Response · Authors · 2024-08-12
> > > **Thanks for the response**
> > >
> > > We are glad that we address your concern about ‘other figures besides 3’ and ‘difference from reference [1,2]’ and we will make sure to add these discussions to the revised version. We will also incorporate the statement you mentioned into the main body of the paper to highlight our contribution, which we believe will provide valuable insights for future work.
> > >
> > > We have addressed the ‘concern about data overlap’ in our response to all reviewers. In summary, there is no overlap between the training data and the test data.
> > >
> > > Thank you once again for your insightful questions. Below is our response to the rest of your questions:
> > >
> > > ***Q1: Concerning memorization vs computation/recombination***
> > >
> > > We acknowledge that it is partly philosophical and difficult to prove that "direct output" is a requirement for memorization. However, as noted in other memorization tasks mentioned in our previous response, where the model uses only a few layers to make the prediction, we believe it is reasonable to hypothesize that the model is performing computation rather than memorization. In later sections of our paper, we correctly split the dataset to fine-tune the model. Through ablation studies and Fourier analysis, we provide additional evidence to support the claim that the model is indeed computing rather than memorizing.
> > >
> > > ***Q2: superposition phenomenon***
> > >
> > > Thank you for your comments. We would like to clarify that the purpose of Figure 4 is to help readers understand how the results in Fourier space correspond to number space. Our experiments are primarily conducted in Fourier space, which may cause some confusion regarding how the results in Fourier space enable the model to identify the correct answer in number space. As explained in lines 137-150, we aim to help readers grasp, for example, why we assert that the Fourier component with a period of 2 is performing a ‘mod 2’ task. Although the superposition phenomenon is not the primary focus of our contribution, we believe that studying this phenomenon (using the top-k Fourier components to express the final logits) is an interesting direction. Since with only a few components, the model can accurately express the final prediction, a potential direction for future research could be making the model to process only with these outlier Fourier components to enhance LLMs’ speed and performance on arithmetic tasks.
> > >
> > > ***Q3: "primarily performed by the MLP modules alone," is inaccurate***
> > >
> > > Thanks for your feedback! We acknowledge that the word ‘alone’ is misleading. We have changed the sentence to ‘As shown in Table 1, the approximation tasks are primarily performed by the MLP modules, with contributions from the attention modules as well.’
> > >
> > > Thanks for reading our responses! Please let us know if the above addresses the questions and concerns.
> > >
> > >
> > > Sincerely,
> > >
> > > Authors

---

> ### Author Response · Authors · 2024-08-06
> **More details about Q5 and Q6**
>
> ***Q5: Figure 8 (4-shot) and figure 15 (multiplication), figure 17 (other format), and figure 18 (GPT-J) are not as clean as Figure 3***
>
> **Figure 17 (alternative format)**: Compared to Figure 3, Figure 17 also clearly shows the outlier frequency components. However, the outlier components in Figure 17 are not distributed across as many layers as in Figure 3, especially for the attention output. From the color bars, we can see that in Figure 17, the magnitude for the 10-period component in the last few layers is larger, particularly for the attention output. This suggests that the last few layers in Figure 17 learn the ‘mod 10’ task well and contribute more to the final logits compared to other layers. Hence, if we clip the values above 600 to 600, we can achieve results similar to those in Figure 3.
>
> **Figures 8 (4-shot) and 18 (GPT-J)**: Compared to Figure 3, we can see the same outlier components in Figures 8 and 18, but they are not as clean as in Figure 3. Our experiments show that the accuracy for the 4-shot Phi2 (Figure 8) is 55.44%, and the accuracy for 4-shot GPT-J (Figure 18) is 72.21%. Hence, we believe that without fine-tuning, the model does not fully leverage Fourier features to solve the addition problem. Even though the pre-trained model has learned to represent the numbers in Fourier space, it fails to fully understand the question phrasing and apply the Fourier method to solve it. We believe that better leveraging the Fourier features to improve the models’ performance on arithmetic tasks can be an interesting direction.
>
> **Figure 15 (multiplication)**: We can clearly see the outlier components in Figure 15, but they are not as clean as Figure 3. Note that for multiplication tasks, even if we expand the dataset size for multiplication, the accuracy only reaches 74.58% after fine-tuning. Multiplication is more difficult for LLMs, as shown in [1,2]. Hence, we believe that since the model does not fully learn to leverage Fourier features to solve the problem, Figure 15 appears less clean compared to Figure 3.
>
> [1] McLeish, Sean, et al. "Transformers Can Do Arithmetic with the Right Embeddings." arXiv preprint arXiv:2405.17399 (2024).
>
> [2] Dziri, Nouha, et al. "Faith and fate: Limits of transformers on compositionality." Advances in Neural Information Processing Systems 36 (2024).
>
> ***Q6: Clarify the difference from Nanda et al. (2023) and add Stolfo et al. (2023) to related work.***
>
> Mechanisms of pre-trained LMs …Furthermore, [1] helps to understand the mechanism of pre-trained LMs when solving arithmetic tasks, by illustrating how LMs transmit and process query-relevant information using attention mechanisms and MLP modules.
>
> Fourier features in Neural Networks. …[2] notes that Fourier features occur in shallow transformer models trained on modular addition. In contrast, our work focuses on the addition task for pre-trained LLMs. We specifically explore the necessity of various Fourier components and their contributions to the predictions...
>
> [1] Stolfo, Alessandro, Yonatan Belinkov, and Mrinmaya Sachan. "A mechanistic interpretation of arithmetic reasoning in language models using causal mediation analysis." arXiv preprint arXiv:2305.15054 (2023).
>
> [2]Nanda, Neel, et al. "Progress measures for grokking via mechanistic interpretability." arXiv preprint arXiv:2301.05217 (2023).

---

### Official Review · Reviewer_YRn2 · 2024-07-16

**Soundness:** 3
**Presentation:** 4
**Contribution:** 3
**Rating:** 7
**Confidence:** 3

**Summary:**

This paper analyzes how language models conduct the task of mathematical addition. Specifically, the paper shows that when performing the mathematical addition task, the internal states of language models exhibit periodical patterns, referred to as "Fourier Features" in the paper. The paper then conducts Fourier transformations to analyze such features and shows that (1) high-frequency features are more helpful in reaching the correct prediction (2) filtering out Fourier components causes performance degradation, hence these features are causally important for model predictions (3) such feature arises during pre-training, in the token embedding components. Finally, the paper shows that their findings generalize to other pre-trained LLMs (GPT-J, Phi-2, GPT-4, etc.) since they exhibit similar behaviors.

**Strengths:**

1. This is an in-depth analysis of one specific facade of the internal workings of the LLM.
2. While this does not seem like the first time Fourier features are studied in neural networks, I still find the analysis method quite interesting.
3. The writing quality is high. Findings are very nicely presented in a progressive and easy-to-follow way.

**Weaknesses:**

1. While the study is thorough, I find the scope to be quite narrow -- it's only focused on one specific arithmetic task. I'm skeptical about how much this will help downstream model development.
2. To make the study feasible, the paper focused on the single-token case, which in turn limits the range of addition to <520. This makes the setup very artificial. That said, this is unfortunately an inherent weakness of many similar analysis papers, and I realize it's tricky to reach a good solution.

**Questions:**

### Presentation Suggestions

- Section 3: I know this is mentioned in the existing literature but it would be nice to define "modular addition" before you use it here for the first time.
- L91: I'm confused by $[L]$. Do you mean a range $[0, L)$ or something similar?
- L114: "approximately" is a weird word in this context. I'm not sure what you are trying to say, but to me, it makes perfect sense even if you remove it (similar occurrence in L121).

### Questions

1. How much do you think the choice of positional embedding has played in the creation of Fourier features? To be more specific, if we use learned positional embedding rather than sinusoidal/rotary positional embedding, do you think you'll still observe these features?
2. I might have missed it but I don't think there's ever an explanation for why attention only learns high-frequency features, while MLP has both. Do you have any insights?
3. More of a suggestion: Did you look at stronger open-source models such as the Mixtral ones? It's a pity that you had to do in-context learning for the model to be able to conduct arithmetic addition.

**Limitations:**

The authors should address the limitations their experiment configurations have created, including but not limited to:

- single token case only
- small number addition only
- findings are constrained to a single arithmetic operation, and it's not clear whether they'll generalize to others

(To clarify, not asking for extra experiments, but just acknowledging the limitations)

---

> ### Author Rebuttal · Authors · 2024-08-06
>
> We appreciate the reviewer’s supportive comments and constructive suggestions.
>
> **Q1: The term "modular addition" should be defined before its first use.**
>
> We have added a definition of "modular addition" in Section 3.1 to the revised version of our paper for clarity: "Modular addition is an arithmetic operation in which the sum of two numbers is divided by a modulus, and the remainder of this division is taken as the result. This operation ensures that the resulting value always falls within the range from zero to one less than the modulus, effectively "wrapping around" to zero once the modulus is reached."
>
> **Q2: Confusion about the notation [L].**
>
> $[L]$ stands for $\{1,2,3,4….L\}$. Hence, $h^{(\ell)}$ where $\ell \in [L]$,  refers to one specific hidden state on the residual stream at layer $\ell$. We have included this definition in Section 3.1 in the revised version.
>
> **Q3: The word "approximately" is potentially confusing in the context it's used.**
>
> For line 114, we agree that the term "approximately" might cause confusion. We originally included it to indicate that while outlier components have significantly larger magnitudes, there are still small magnitudes for other components. However, we recognize that this might not be necessary for clarity in this context. Therefore, we have removed "approximately" from lines 114 and 121 to improve clarity.
>
>
> **Q4: How much do you think the choice of positional embedding has influenced the creation of Fourier features?**
>
> In our research, we treat each number as a distinct token with a corresponding token embedding. Consequently, the choice of positional embedding does not directly impact our results and observations. Moreover, it is important to note that the pre-trained embeddings have already shown Fourier features. We believe that when it comes to multiple tokens, these features will also enhance model accuracy. However, this may introduce additional complexity due to the increased number of tokens involved.
>
> **Q5: Why do attention mechanisms tend to learn high-frequency features, whereas MLPs capture both high and low-frequency features?**
>
> Thank you for your insightful question. While our paper focuses primarily on the interpretability of pre-trained LLMs, we can provide insights into this observation based on existing research and theoretical understandings.
>
>    - **The key to do modular addition is to compute the trigonometric functions.** As shown in [2], to solve modular addition $a + b \mod p$, the Transformer model tends to address the equivalent problem $\arg\max_{c} \cos(2\pi(a + b - c)/p)$. The token embedding in transformers represents inputs as $\cos(wa)$, $\cos(wb)$, $\sin(wa)$, and $\sin(wb)$ with different frequencies $w$. Using trigonometric functions, the model computes $\cos(w(a + b))$ and $\sin(w(a + b))$, which are then processed to get $\cos(w(a + b - c))$ and $\sin(w(a + b - c))$.
>   - **Attention can compute trigonometric functions.** Assuming the key and query matrices as identity matrices for simplicity, the attention mechanism can be simplified as $ \text{softmax}(XX^\top)V $. As $\cos(wx)$ and $\sin(wx)$ have been encoded in $X$, the attention can effectively compute the result of $\cos(w_1 a) \cdot \cos(w_2 b)$ with different frequencies $w_1, w_2$. More evidence can be found in [1] Section 5.2. These shows why attention mechanisms are suitable to compute trigonometric functions.
>
> - **MLP can approximate magnitude.**    According to [3], during arithmetic tasks, the information about numbers $a$ and $b$ is processed by MLPs in the later layers of a neural network. If the magnitudes of $a$ and $b$ are represented in specific entries of the hidden state, say $h[i]$ and $h[j]$, an MLP can compute $w[i]h[i] + w[j]h[j]$, which approximates the magnitude of $a + b$. This demonstrates that MLPs can approximate the magnitude of $a+b$.
>
> In summary, the tendency of attention mechanisms to learn modular addition (high-frequency features) while MLPs primarily learn approximation (low-frequency features) can be attributed to their different structures. We hope these insights help clarify the observation and provide a deeper understanding of the underlying mechanisms.
>
> **References:**
>
> 1. Gu, Jiuxiang, et al. "Fourier circuits in neural networks: Unlocking the potential of large language models in mathematical reasoning and modular arithmetic." arXiv preprint arXiv:2402.09469 (2024).
> 2. Nanda, Neel, et al. "Progress measures for grokking via mechanistic interpretability." arXiv preprint arXiv:2301.05217 (2023).
> 3. Stolfo, Alessandro, Yonatan Belinkov, and Mrinmaya Sachan. "A mechanistic interpretation of arithmetic reasoning in language models using causal mediation analysis." arXiv preprint arXiv:2305.15054 (2023).
>
>
> **Q6: Consider using stronger open-source models like Mixtral for arithmetic tasks.**
>
> Thank you for your feedback. We have indeed tried our experiments on stronger open-source models. However, we found that the predictions were not controllable without in-context learning. The models sometimes generated intermediate steps or repeated the questions even when we appended ‘Answer is’ at the end of our prompts. In-context learning helped ensure that the model understood the task and provided the correct answers. We believe that in-context learning does not change the model’s underlying strategy to solve the tasks but rather helps the model to respond appropriately.Additionally, we emphasize that for closed-source models, there is also evidence supporting the existence of Fourier features. This further validates our findings and suggests that the observed phenomena are not limited to the specific models we tested.

---

### Author Rebuttal · Authors · 2024-08-06

**Response to All Reviewers:**

We thank reviewers [R1(YRn2), R2(XwiY), R3(wYe3), R4(mHEL)] for their thoughtful and highly supportive feedback! We are glad that the reviewers found the analysis method interesting and  [R1, R3], the observations about the application of Fourier features in language models performing addition tasks insightful and novel [R2, R4], the presentation of our findings easy to follow [R1, R2], and the experimental results compelling, especially regarding the roles of Fourier components in model behavior and the importance of model pretraining [R3, R4].


***Q1: Limitations pointed out by the reviewers: focusing on a single arithmetic task (addition) and the single-token case limiting the range to numbers less than 520.***

1. **Scope of Study**: As an interpretability paper, we focused on a single-token case with numbers less than 520 to provide a controlled and clear analysis environment. This setup allows us to dive deep into understanding the fundamental mechanisms at play. It is a starting point to inspire further research and improvements in the model’s performance on arithmetic tasks. These limitations are inherent to many interpretability studies. For example, [1,2] focus on modular addition, a simpler task than general addition, and their analyses are conducted on shallow neural networks or transformers. In contrast, our study investigates pre-trained LLMs on the addition task, offering insights into more complex and realistic scenarios.

2. **Broader Implications**: We emphasize that the observations on pre-trained number embeddings are not limited to addition tasks. The number embeddings after pre-training are sparse in the Fourier space for many pre-trained LLMs without fine-tuning (Figure 14). This suggests that the model does not learn to use Fourier features to embed numbers just for one task, implying that these learned Fourier features will benefit other number-related tasks as well. In this paper, we focus on addition to help readers understand how these Fourier features assist the model in solving addition problems. We also provide evidence that the model uses these Fourier features in multiplication (Figure 15, Section C.2).

3. **Future Research Directions**: The insights and observations from this paper can inspire many interesting future directions:
    - Exploring the potential of adding a regularizer to help models learn high-frequency Fourier features to enhance performance on number-related tasks.
    - Investigating the training dynamics of how models learn to use Fourier features to represent numbers and solve arithmetic problems during pre-training.
    - Understanding why models tend to use these Fourier features to solve problems.
    - Examining whether models still utilize these Fourier features when dealing with numbers tokenized using sub-word tokenizers.
    - Investigating the use of digit-wise tokenizers, such as in Llama, to see if the model leverages Fourier features for arithmetic tasks.
    - Developing strategies to help the model learn Fourier features in both cases mentioned above to improve performance on arithmetic tasks.
    - Improving the model's performance on larger numbers or decimal numbers by finding strategies to correctly embed numbers.

By highlighting these points, we hope to demonstrate the broader implications of our findings and the potential for future research inspired by our work.


**References:**

1. Morwani, Depen, et al. "Feature emergence via margin maximization: case studies in algebraic tasks." arXiv preprint arXiv:2311.07568 (2023).
2. Nanda, Neel, et al. "Progress measures for grokking via mechanistic interpretability." arXiv preprint arXiv:2301.05217 (2023).


***Q2: Concern about data overlap in training and test sets due to the use of the same mathematical expressions with different templates.***

We apologize for the confusion. In line 595, the mention of generating "various phrasings" for addition questions was a typo. In the actual experiments, we used a distinct phrasing for each pair of numbers, selecting one template from five available templates. This ensures that every unique pair of numbers between 0 and 260 is presented with a consistent phrasing from these templates. We have fixed that typo in the revised version.

---

> ### Comment · Reviewer_XwiY · 2024-08-12
> **Clarify Q2 about data overlap**
>
> Can you clarify the answer about Q2 above? If there is a pair of numbers x, y, between 0 and 260, which is in the training data, could the same pair x, y appear also in the test data (even with a different phrasing) or not?

---

> > ### Author Response · Authors · 2024-08-12
> >
> > Thank you for your question. To clarify, no, the same pair of numbers $x,y$ that appears in the training data does not appear in the test data, even with a different phrasing.
> >
> > Best,
> >
> > Authors

---

> > > ### Comment · Reviewer_XwiY · 2024-08-12
> > > **Thanks for clarifying!**
> > >
> > > thanks!

---

### Decision · Program_Chairs · 2024-09-25

**Decision:**

Accept (poster)

**Comment:**

The paper analyzes how LMs perform addition using Fourier features and with logit lens as well. The work makes a solid niche contribution in our understanding of LMs mechanism of computing addition. While the study is conducted on a very small dataset and mainly the addition property of LMs is analyzed, the work is solid and it contributes to own understanding of LMs.